# A Physics-Inspired Optimizer: Velocity Regularized Adam

**Pranav Vaidhyanathan**[1,3][*] **Lucas Schorling**[1,5][*] **Natalia Ares**[1,5] **Michael A. Osborne**[1,3][†]

[1] University of Oxford, United Kingdom
[3]{pranav, mosb}@robots.ox.ac.uk
[5]{lucas.schorling, natalia.ares}@eng.ox.ac.uk

## Abstract

We introduce Velocity-Regularized Adam (VRAdam), a physics-inspired optimizer for training deep neural networks that draws on ideas from quartic terms for kinetic energy with its stabilizing effects on various system dynamics. Previous algorithms, including the ubiquitous Adam, operate at the so-called adaptive edge of stability regime during training, leading to rapid oscillations and slowed convergence of loss. However, VRAdam adds a higher order penalty on the learning rate based on the velocity such that the algorithm automatically slows down whenever weight updates become large. In practice, we observe that the effective dynamic learning rate shrinks in high-velocity regimes, and damping oscillations. By combining this velocity-based regularizer for global damping with Adam's per-parameter scaling, we create a powerful hybrid optimizer. For this optimizer, we provide rigorous theoretical analysis of operation at the edge of stability from a physical and control perspective for the momentum. Furthermore, we derive convergence bounds with the rate $\mathcal{O}(\ln(N)/\sqrt{N})$ for a stochastic non-convex objective under mild assumptions. We demonstrate that VRAdam exceeds the performance of standard optimizers including AdamW. We benchmark various tasks such as image classification, language modeling, and generative modeling using diverse architectures and training methodologies including Convolutional Neural Networks (CNNs), Transformers, and GFlowNets.

## 1 Introduction

Optimizing the parameters of deep neural networks remains a cornerstone of progress in machine learning. Improving on the core idea of Stochastic Gradient Descent (SGD) (Sutskever et al., 2013), adaptive methods like Adam (Adaptive Moment Estimation) (Kingma & Ba, 2015) have become ubiquitous due to their practical effectiveness across diverse tasks and architectures. Despite its success, the performance of Adam can be sensitive to hyperparameter choices and its training dynamics can exhibit instabilities (Reddi et al., 2019). Furthermore, fully understanding the training dynamics of deep neural networks remains an open challenge (Wang & Choromanska, 2025), and even small improvements to existing optimization algorithms can often lead to significant reductions in resource consumption.

One line of work, observed empirically, is that training often occurs at the edge of stability (Cohen et al., 2022; 2024), a regime for which the largest eigenvalue, also called sharpness, of the loss Hessian equilibrates around a fixed value proportional to the inverse of the learning rate (LR). This seems in contrast with common presumptions in classical optimization theory and has profound implications for convergence speed, stability, and generalization. In classical optimization, higher LRs lead to faster convergence at the cost of oscillations or divergence if stability constraints (depending on the loss landscape) are violated (Boyd & Vandenberghe, 2004). This pathological behavior of optimizers, like AdamW, leads to instabilities and a slowed decrease of the loss.

---

[*]Equal Contribution
[†]Corresponding author

These challenges motivate the exploration of alternative optimization strategies. In line with the origins of machine learning itself (Hopfield, 1982; Ackley et al., 1985), one promising avenue draws inspiration from physics. For that, the optimization trajectory is conceptualized as a discretized motion of a particle within the high-dimensional loss landscape. Both from the structure of the "potential" landscape and the discretization, instabilities may arise from excessive "velocity" or overly large step sizes. This perspective suggests that mechanisms from high-energy and non-classical physics applied to optimization can improve aspects of stability. Building upon the established success of momentum, which incorporates velocity into gradient updates, recent ideas have explored a maximal velocity, drawing parallels to the speed of light in the theory of special relativity (França et al., 2020).

This work introduces a class of physics-inspired optimizers, termed Velocity-Regularized Adam (VRAdam), designed to improve upon the stability and performance of standard adaptive methods (Kingma & Ba, 2015; Loshchilov & Hutter, 2017). Inspired by quartic terms used to model kinetic energy in more stable systems such as classical time crystals (Shapere & Wilczek, 2012) and heavy quark modeling using non-relativistic quantum chromodynamics (NRQCD) (Braaten, 1997) known for their unique stability properties, VRAdam adapts this as a heuristic and introduces a novel regularization mechanism. This mechanism controls the effective learning rate $\eta$ via penalizing high velocity, namely $\eta_t = \alpha_0/(1 + \min(\beta_3||v_t||^2, \alpha_1))$.

Equipped with this new optimizer, VRAdam, we probe its dynamics at the adaptive edge of stability, observe faster convergence, and analyze its sharpness empirically against AdamW and Sharpness Aware Minimization (SAM) (Foret et al., 2021). We also introduce rigorous theoretical analysis of the global uniform exponential stability of momentum (Weber et al., 2024) as well as a physics-inspired Lyapunov candidate derived from our Lagrangian that demonstrates stability properties. With well-tuned hyperparameters, we benchmark VRAdam against AdamW, RAdam (Liu et al., 2020), SGD with Nesterov momentum and RMSProp (Ruder, 2017) on image classification with the CIFAR-10 dataset and the convolutional neural network architecture, on language modeling with transformers on the WikiText2 dataset, and a generative modeling task with GFlowNets and report improved performance on all tasks. We also report increased performance against AdamW on large scale training for language models such as GPT (Brown et al., 2020) with marginal computational increase in overhead.

With this work, we contribute:

- VRAdam, a physics-inspired and interpretable modification to AdamW,

- Adaptive edge of stability analysis of VRAdam with faster convergence and associated empirical and theoretical evidence from momentum analysis,

- Convergence bound for non-convex stochastic objective,

- VRAdam outperforms AdamW and other optimizers on a wide range of benchmarks.

## 2 BACKGROUND

**Edge of stability.** The training of deep neural networks does not follow classical optimization trajectories when trained with full-batch gradient descent (GD). However, during training, these models experience a surprising phase called the edge of stability (EoS). In this phase, the loss Hessian's largest eigenvalue ($\lambda_{\max}$) rises to approximately $2/\eta$, the numerical stability limit determined by the learning rate $\eta$. At EoS, the eigenvalue persists at this threshold, causing short-term, non-monotonic oscillations in the loss function. Despite these oscillations, the model still achieves long-term descent in the loss, though at the cost of slower convergence (Arora et al., 2022; Cohen et al., 2022).

More recently, empirical bounds on the adaptive edge of stability (AEoS) have been observed for adaptive optimizers such as Adam (Cohen et al., 2024). Here, the relevant stability threshold involves preconditioning the Hessian $H_t$, where the precondition is constructed from the exponential moving average (EMA) of past element-wise squared gradients $m_t$:

$$P_t^{-1}H_t, \quad P_t = \mathrm{diag}\left(\sqrt{m_t} + \varepsilon\right), \tag{1}$$

This adaptive preconditioning coincides with the learning rate scaling in Adam (compare Alg. 1) and scales down the step size in high-variance (typically high-curvature) directions as well as scales up in low-variance ones. Since the local stability of an optimizer around a minimizer depends on the eigenvalues of the quadratic Taylor approximation $L(x) \approx \frac{1}{2}x^\top H x$, Adam's dynamics are shown to be stable (Cohen et al., 2024) as long as

$$\lambda_{\max}\left(P_t^{-1} H_t\right) < \frac{2 + 2\beta_1}{(1 - \beta_1)\,\eta} = \frac{38}{\eta} \quad (\beta_1 = 0.9)\,. \tag{2}$$

However, as this threshold is attained, the adaptive oscillatory regime can slow final convergence, as the optimizer continually adjusts its preconditioner to maintain stability as well (Song & Yun, 2023).

**Physical origins of exotic Lagrangians.** To better comprehend and navigate the edge of stability regime, we can draw inspiration from the physics governing complex optimization scenarios. The deep insights provided by the interplay of physics and machine learning frameworks have been demonstrated in various scenarios, such as the improved interpretation of Neural Tangent Kernels (NTK) through Langevin dynamics (Avidan et al., 2025). One central concept in physics is the Lagrangian $\mathcal{L}(x, v) = T(v) - V(x)$ of a system, which is a function of position $x$ and velocity $v$ and (typically) defined as the difference between kinetic energy $T$ and potential energy $V$ from which the equation of motion can be derived via the Euler-Lagrange equation. In this work, we investigate non-standard Lagrangian formulations of physical phenomena with excellent stability conditions. For example, the stability of seemingly disparate quantum systems like heavy quarkonia (described by NRQCD) and classical time crystals share conceptual parallels rooted in higher-order velocity terms. These terms fundamentally reshape energy landscapes by creating non-standard dispersion relations, establishing invariant submanifolds in phase space where stable configurations emerge as attractors or limit cycles. Further information regarding these systems are found in Appendix A.

## 3 METHOD

To translate this physics insight into an optimizer design, we identify the stabilizing aspects of such phenomena such as the heavy-quark momentum with the optimizer's global momentum buffer $v$. Accordingly, we posit a kinetic energy of the form: $T_{\mathrm{VRAdam}}(v) = \frac{m}{2}\|v\|^2 + \frac{\beta_3}{4}\|v\|^4$, where $m$ is the mass and $\beta_3$ is a tunable parameter. The Lagrangian then becomes

$$\mathcal{L}(x, v) = \frac{m}{2}v^2 + \frac{\beta_3}{4}v^4 - V(x), \tag{3}$$

for which we solve the Euler-Lagrange equation $\frac{d}{dt}\frac{\partial \mathcal{L}}{\partial v} - \frac{\partial \mathcal{L}}{\partial x} = 0$. We know that the loss of a neural network with parameters $x$ can be thought of as the potential landscape (Holderrieth et al., 2024), such that $\frac{\partial \mathcal{L}}{\partial x} = -\frac{\partial V(x)}{\partial x} = -\nabla L_{\mathrm{loss}}(x)$.

The Euler-Lagrange equation then becomes

$$\frac{d}{dt}\left[\left(m + \beta_3\|v\|^2\right) v\right] = -\nabla L_{\mathrm{Loss}}(x), \tag{4}$$

which can be rearranged to

$$\dot{v} = -\nabla L_{\mathrm{Loss}}(x)/(m + 3\beta_3\|v\|^2), \dot{x} = v \tag{5}$$

where the dot corresponds to the time derivative. Rather than explicitly constructing an optimizer based on an ordinary differential equation solver for Eq. 5 (via discretization and introduction of dissipation, e.g. França et al. (2020)), we utilize the term, $(1/(m + 3\beta_3\|v\|^2)$, and embed it as a dynamic learning rate into AdamW, to enhance the successfully proven properties. Note that this expression from Eq. 5 is derived for the 1-D case, where the dynamics $\dot{v}$ is collinear with $v$. In particular, this approach avoids choosing specific forms associated with various different integrators. The full velocity regularized Adam is given in Alg. 1, where we highlight in blue the changes to AdamW.

In Fig. 1, the vector field in Eq. 5 is plotted for the case of the loss function being a simple quadratic. Compared to kinetic energy term without quartic velocity, the vector field is "squeezed"

in $v$ direction, and the resulting trajectories are not circular. In this idealized setting, we can visualize the performance of VRAdam and Adam simplified to VRMomentum and Momentum, which corresponds to Alg. 1 without second-order moment estimates or bias corrections (setting $m_t = 1$ and dropping lines 6, 8, and 9). After the first step, VRMomentum has a lower step size than Momentum, which leads to fewer oscillations. Through reparameterization and modification, we obtain

**Algorithm 1** VRAdam optimizer. $f(\theta)$: objective function; $\beta_1, \beta_2 \in [0, 1)$; $v_t$: velocity estimate; $m_t$: second-moment estimate; $\eta_t$: dynamic learning rate at step $t$; $\alpha_0$: maximal learning rate; $\alpha_0/(1 + \alpha_1)$: minimal learning rate; $\beta_3$: velocity penalizer. Code available at Vaidhyanathan et al. (2026).

1: **Input:** $f(\theta), \theta_0, \alpha_0, \alpha_1, \beta_1, \beta_2, \beta_3,$ $\epsilon, \lambda$
2: Initialize $v_0 \leftarrow 0, m_0 \leftarrow 0$
3: **for** $t = 1, \ldots, T$ **do**
4:     $g_t \leftarrow \nabla f(\theta_{t-1})$
5:     $v_t \leftarrow \beta_1 v_{t-1} + (1 - \beta_1) g_t$
6:     $m_t \leftarrow \beta_2 m_{t-1} + (1 - \beta_2) g_t^2$
7:     $\eta_t \leftarrow \alpha_0/(1 + \min(\beta_3||v_t||^2, \alpha_1))$

8:     $\hat{v}_t \leftarrow v_t/(1 - \beta_1^t)$
9:     $\hat{m}_t \leftarrow m_t/(1 - \beta_2^t)$
10:    $\theta_t \leftarrow \theta_{t-1}(1 - \eta_t \lambda) - \eta_t \dfrac{\hat{v}_t}{\sqrt{\hat{m}_t} + \epsilon}$
11: **end for**
12: **Output:** $\theta_T$

Figure 1: Vector field $\dot{v} = -x/(1 + 3v^2)$ and $\dot{x} = v$ derived by solving the Euler–Lagrange equation for $\mathcal{L} = v^2/2 + v^4/4 - x^2/2$. Black lines are continuous trajectories; blue/orange show VRMomentum vs. Momentum steps.

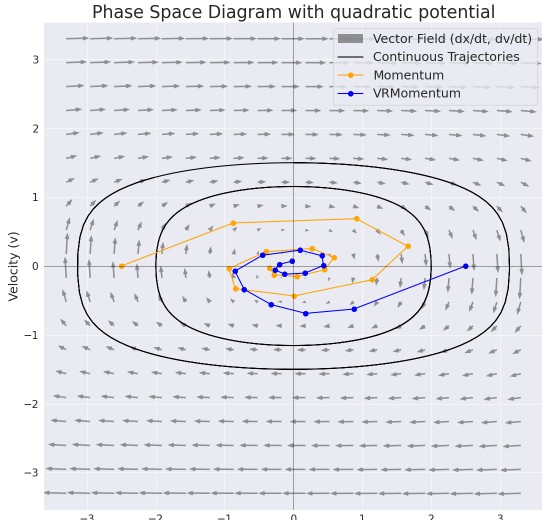

the dynamic learning rate $\eta_t = \alpha_0/(1 + \min(\beta_3||v_t||^2, \alpha_1))$ for timestep $t$ for VRAdam, where $\alpha_0$ and $\alpha_1$ control the maximal and minimal LR respectively, and $\beta_3$ controls the strength of the velocity penalty. This is inspired by the bound introduced to $v^2$ in physical setting as discussed in Appendix A. The parameterization of LR, compared to the physically derived one, clips the velocity to avoid getting stuck if gradients and therefore velocity become large. Weight decay is applied in the traditional manner. An implementation of VRAdam is available at Vaidhyanathan et al. (2026).

## 4 ANALYSIS

### 4.1 EMPIRICAL ANALYSIS

For this analysis, following Cohen et al. (2024), we train a ResNet-32 architecture on CIFAR-10 for an image classification task, with VRAdam, Adam and SAM. The training is stopped as soon as the training loss falls below 0.1 or an accuracy of 0.97 is reached. In Fig. 2 (a) and (b), the training curves of VRAdam indicate faster convergence in both minimizing training loss as well as maximizing training accuracy as compared to Adam and SAM, known for sharpness minimization. When juxtaposed with the sharpness comparison depicted in Fig. 2 (c), we observe that the maximum eigenvalue (sharpness) of the preconditioned Hessian of the loss, remains adaptable to faster convergence due to the dynamic learning rate adjustments induced by VRAdam.

The effective learning rate of VRAdam is shown in Fig. 2 (d). During the first 25 iterations, the learning rate dynamically decreases before increasing close to the maximal value allowed LR as we converge to the minima. The bound on the minimal LR is not active in this example, while the base LR of Adam stays constant throughout training. As described in the seminal work of Schmidhuber *et al.*, we note that minima with lower sharpness are associated with better generalization (Hochreiter & Schmidhuber, 1997; Foret et al., 2021). We can also observe, that VRAdam's dynamic learning rate quickly moves to the maximal LR to exploit the loss landscape optimally, while exploiting the trade-off between the adaptive edge of stability and faster convergence.

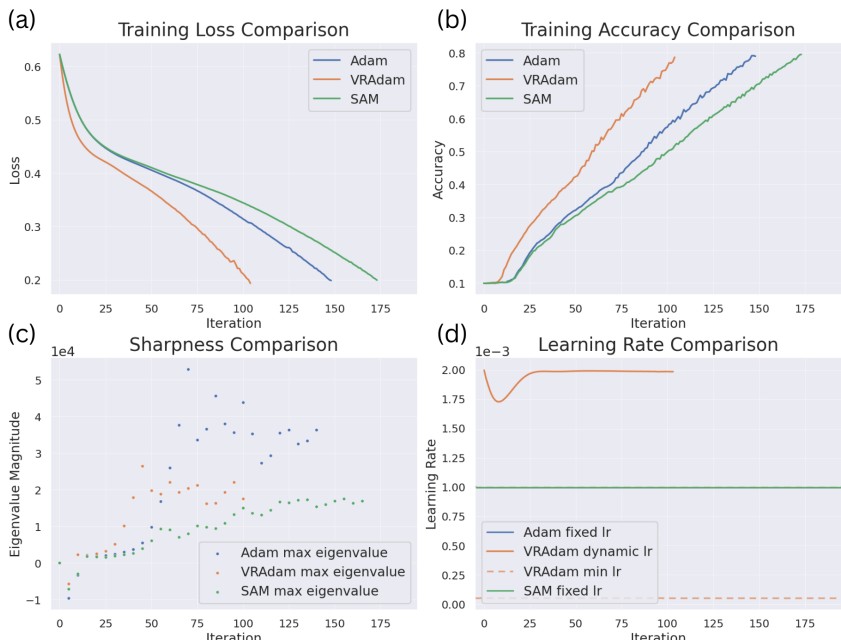

Figure 2: **(a)** Training loss curves for VRAdam, Adam, and SAM (Foret et al., 2021) of ResNet-32 on CIFAR-10 **(b)** training accuracy curves **(c)** plot of maximal eigenvalues of the loss Hessian **(d)** effective learning rate during training. Hyperparameters for these plots are provided in Appendix E.4.

## 4.2 STABILITY OF VRADAM

We analyze the behavior of VRAdam in the adaptive edge of stability regime compared to that of Adam in a momentum ablation setting without parameter-scaling based on second order moments estimates, bias corrections or weight decay. By simply adding weight decay, the admissible base step is constrained, the rest of the proof remains as follows. We first give a quadratic *warm-up* (Theorem 4.1) as the main result of this section; immediately after, we state a nonconvex corollary that replaces the global convexity assumption by a trajectory-level curvature bound together with an analytical shift (decoupled weight decay).

**Objective function with minimal curvature assumption.** Along the realized trajectory, the Hessian satisfies a two-sided spectral bound

$$-m\,I \;\preceq\; \nabla^2 f(\theta_t) \;\preceq\; L\,I \qquad \text{for all iterates } \theta_t, \tag{6}$$

for some $m \geq 0$, $L > 0$. We include decoupled weight decay of strength $\lambda \geq 0$ in the update (Alg. 1, line 10). As a standard simplification, we analyze first the quadratic model

$$f(\theta) = \frac{1}{2}(\theta - \theta^*)^\top H(\theta - \theta^*) \tag{7}$$

where $H$ is a positive definite Hessian matrix. Let $x_t = \theta_t - \theta^*$ be the error. The gradient is, $g_t = \nabla f(\theta_{t-1}) = Hx_{t-1}$. Let $\mu = \lambda_{\min}(H)$ and $L := \lambda_{\max}(H)$. From Algorithm 1, we know that $\eta_t \in \mathcal{H} := [\eta_{\min}, \alpha_0]$ and $\eta_{\min} = \alpha_0/(1+\alpha_1) > 0$. Let $\beta_1$ from Algorithm 1 be $\beta$ for this analysis. The following theorem and proof provides sufficient but not necessary conditions for stability in discrete time. Thus proving a more restrictive case of the optimizer.

**Theorem 4.1** (Uniform exponential stability of VRMomentum). *Consider $f(\theta)$ with $0 \prec H \preceq LI$. Let VRMomentum be $\beta \in [0,1)$, $\beta_3 > 0$, $\alpha_0 > 0$, $\alpha_1 \in (0,\infty]$, and set $\lambda = 0$ in this warm-up (no weight decay in the matrix recursion). If $\alpha_0 L < B(\beta) = \frac{2(1+\beta)}{1-\beta}$ or $\eta_{\min}L < B(\beta)$ if the LR clip is active, then for any realization of $\{\eta_t\}$ generated by the gate $\eta_t = \alpha_0/(1 + \min(\beta_3\|v_t\|^2, \alpha_1))$, the origin is a globally uniformly exponentially stable equilibrium. Moreover, there exists a Common Quadratic Lyapunov Function (CQLF), $V(z) = z^\top P z$ with $P \succ 0$ such that $V(z_t) \leq (1-\epsilon)V(z_{t-1})$ for some $\epsilon \in (0,1)$.*

*Proof.* Define the 2d-state $z_t = (x_t, v_t)$ and the parameterized update matrix

$$A(\eta) := \begin{pmatrix} I - \eta(1-\beta)H & -\eta\beta I \\ (1-\beta)H & \beta I \end{pmatrix}, \quad z_t = A(\eta_t)z_{t-1}. \tag{8}$$

Let $H = Q^\top \mathrm{diag}(h_i)Q$ with $0 < \mu \le h_i \le L$. In the eigenbasis of $H$, the dynamics split into $d$ identical $2 \times 2$ subsystems that share the same scalar $\eta_t$ and differ only by the curvature $h \in [\mu, L]$:

$$\begin{pmatrix} \xi_t \\ v_t \end{pmatrix} = A_h(\eta_t) \begin{pmatrix} \xi_{t-1} \\ v_{t-1} \end{pmatrix}, \quad A_h(\eta) := \begin{pmatrix} 1 - \eta(1-\beta)h & -\eta\beta \\ (1-\beta)h & \beta \end{pmatrix}. \tag{9}$$

Hence it suffices to build a CQLF for the family $\{A_h(\eta) : h \in [\mu, L], \eta \in \mathcal{H}\}$.

For a fixed $(h, \eta)$, the characteristic polynomial is

$$\lambda^2 - (1 + \beta - \eta(1-\beta)h)\lambda + \beta = 0, \tag{10}$$

so the Schur criterion gives stability iff $\eta(1-\beta)h < 2(1+\beta)$. In particular, if

$$\alpha_0 L < B := \frac{2(1+\beta)}{1-\beta} \tag{11}$$

then every $A_h(\eta)$ with $\eta \in \mathcal{H}$ is Schur-stable. (This is the Adam/AEoS bound specialized to $P = I$ (Cohen et al., 2024))

If, instead of the quadratic model, we assume the trajectory-level curvature range $h \in [-m, L]$ and include decoupled weight decay $\lambda \ge 0$ in the update, the same $2 \times 2$ calculation yields the uniform sufficient condition

$$\lambda > m \quad \text{and} \quad \alpha_0\big((1-\beta)L + (1+\beta)\lambda\big) < 2(1+\beta). \tag{12}$$

Setting $\lambda = 0$ and $m = 0$ recovers eq. 11.

We now produce $P \succ 0$ such that

$$A_h(\eta)^\top P A_h(\eta) - P \preceq -\epsilon I_2 \quad \forall \eta \in \mathcal{H}, \forall h \in [\mu, L], \tag{13}$$

for some $\epsilon > 0$. Take the block-diagonal, curvature-agnostic form

$$P = \mathrm{diag}(p_1 I, p_2 I), \quad p_1 > 0, p_2 > 0, \tag{14}$$

which lifts to $P_d = \mathrm{diag}(p_1 I_d, p_2 I_d)$ in 2d dimensions.

For a fixed $(h, \eta)$, set

$$\Delta(\eta, h) := A_h(\eta)^\top P A_h(\eta) - P = \begin{pmatrix} \Delta_{11} & \Delta_{12} \\ \Delta_{12} & \Delta_{22} \end{pmatrix} \tag{15}$$

with

$$\Delta_{11}(\eta, h) = (1-\beta)h\left[ -2p_1\eta + (1-\beta)h\left(p_1\eta^2 + p_2\right) \right], \tag{16}$$

$$\Delta_{22}(\eta, h) = p_1\eta^2\beta^2 + p_2(\beta^2 - 1), \tag{17}$$

$$\Delta_{12}(\eta, h) = -p_1\eta\beta(1 - \eta(1-\beta)h) + p_2\beta(1-\beta)h. \tag{18}$$

We first force strict negativity on the diagonal, uniformly over $\eta \in \mathcal{H}$ and $h \in [\mu, L]$. Using the bounds $\eta \in [\eta_{\min}, \alpha_0]$ and $h \in [\mu, L]$, a sufficient pair of conditions for $\Delta_{11} \le -\delta_1$ and $\Delta_{22} \le -\delta_2$ is

$$\underbrace{(\alpha_0^2 p_1 + p_2)L}_{\text{upper bound for the } h^2\text{-term}} < \underbrace{\frac{2\eta_{\min}}{1-\beta}p_1}_{\text{lower bound for the linear } h\text{-term}}, \tag{19}$$

$$\underbrace{p_2(1-\beta^2)}_{\text{negative term magnitude}} > \underbrace{p_1\alpha_0^2\beta^2}_{\text{positive term bound}}. \tag{20}$$

(Here we used the worst cases $\eta = \alpha_0$ and $h = L$ wherever they make the expression largest.)

Intersecting eq. 19, 20 with $p_1 = 1$ gives a nonempty interval iff

$$\alpha_0^2 L < 2(1+\beta)\,\eta_{\min} \qquad \Longleftrightarrow \qquad \frac{\alpha_0^2 \beta^2}{1-\beta^2} \; < \; \frac{2\eta_{\min}}{(1-\beta)L} - \alpha_0^2. \tag{21}$$

Indeed, set $p_1 = 1$ and choose any

$$p_2 \in \left( \frac{\alpha_0^2 \beta^2}{1-\beta^2}, \frac{2\eta_{\min}}{(1-\beta)L} - \alpha_0^2 \right). \tag{22}$$

Next, to pass from diagonal negativity to matrix negativity, we use the Schur complement:

$$\Delta \preceq -\epsilon I_2 \iff \Delta_{11} \leq -\epsilon, \quad \Delta_{22} - \frac{\Delta_{12}^2}{\Delta_{11}} \leq -\epsilon. \tag{23}$$

Under eq. 19- 20, $\Delta_{11}$ and $\Delta_{22}$ are uniformly $\leq -\delta$ for some $\delta > 0$. A direct (but routine) bound gives, for all $\eta \in \mathcal{H}, h \in [\mu, L]$,

$$\frac{\Delta_{12}^2}{-\Delta_{11}} \leq \frac{\beta^2 (p_1 \alpha_0 + p_2(1-\beta)L)^2}{2\eta_{\min} p_1 - (\alpha_0^2 p_1 + p_2)(1-\beta)L}, \tag{24}$$

and the right-hand side is strictly smaller than $-\Delta_{22}$ when eq. 19, 20 hold with slack. Thus there exists $\epsilon > 0$ such that eq. 13 is satisfied.

For fixed $h$ and $P \succ 0$, the map

$$\eta \mapsto A_h(\eta)^\top P A_h(\eta) = (A_0 + \eta A_1)^\top P (A_0 + \eta A_1) \tag{25}$$

is matrix-convex in $\eta$ (its second derivative is $2A_1^\top P A_1 \succeq 0$). Therefore, if the inequality

$$A_h(\eta)^\top P A_h(\eta) - P \preceq -\epsilon I_2 \tag{26}$$

holds at the endpoints $\eta = \eta_{\min}$ and $\eta = \alpha_0$, it holds for all $\eta \in [\eta_{\min}, \alpha_0]$. Under eq. 21, the choice eq. 22 does exactly that.

The block choice $P_d = \mathrm{diag}(p_1 I_d, p_2 I_d)$ certifies

$$A(\eta)^\top P_d A(\eta) - P_d \preceq -\epsilon I_{2d} \quad \forall \eta \in \mathcal{H}, \tag{27}$$

hence the quadratic Lyapunov $V(z) = z^\top P_d z$ yields

$$V(z_t) \leq (1-c) V(z_{t-1}), \quad c = \frac{\epsilon}{\lambda_{\max}(P_d)} \in (0, 1), \tag{28}$$

which implies global uniform exponential stability and concludes the proof.

**Corollary for nonconvex stability via analytical shift.** Assume along the VRAdam trajectory that $-mI \preceq \nabla^2 f(\theta_t) \preceq LI$ and include decoupled weight decay $\lambda > m$. If eq. 12 holds (with $\alpha_0$ or $\eta_{\min}$), then every $2 \times 2$ eigen-direction block is Schur-stable uniformly over $h \in [-m, L]$ and $\eta \in \mathcal{H}$, hence the origin is globally uniformly exponentially stable. (Setting $\lambda = 0$ and $m = 0$ reduces to eq. 11.)

In the Appendix B, an alternative Lyapunov candidate based on Lagrangian physics is presented.

### 4.3 GLOBAL VS PER-PARAMETER LR ADAPTIVITY

We contrast two ways of modulating the step magnitude, namely, (i) *per-parameter* control via a time-varying diagonal matrix (element-wise scaling), and (ii) a *global* scalar gate $\eta_t$ that multiplies the entire (preconditioned) step as in Alg. 1. The global gate introduces key advantages in the edge-of-stability regime.

**Uniform Lyapunov stability under arbitrary scalar switching.** In the momentum ablation on a quadratic, the global-gated dynamics decouple into identical $2 \times 2$ blocks $A_h(\eta_t)$ in the eigenbasis of the Hessian (see the construction surrounding eq. 13–eq. 14 and the Schur bound eq. 11). Theorem 4.1 produces a curvature-agnostic Common Quadratic Lyapunov Function $V(z) = z^\top P z$ with $P \succ 0$ independent of $h$ and the time-varying $\eta_t \in [\eta_{\min}, \alpha_0]$, certifying *global uniform* exponential

stability whenever $\alpha_0 L < \frac{2(1+\beta_1)}{1-\beta_1}$ (or the same with $\eta_{\min}$ when the clip is active), i.e., the Schur condition eq. 11. Hence stability holds for *any* scalar gate sequence generated by Alg. 1. (See Sec. 4.2, Thm. 4.1.)

**Global, rotation-invariant control of AEoS with bounded steps.** The velocity-based gate in Alg. 1, derived from the quartic Lagrangian eq. 3 and the collinear 1-D reduction eq. 5, implies a dimension-free bound on the update norm and raises the instantaneous stability threshold whenever the measured velocity grows:

$$\|\theta_t - \theta_{t-1}\| = \eta_t \|v_t\| \le \frac{\alpha_0}{2\sqrt{\beta_3}}, \qquad L_{\text{EoS}}(t) = \frac{2(1+\beta_1)}{(1-\beta_1)\eta_t} = \frac{2(1+\beta_1)}{(1-\beta_1)\alpha_0}\Big(1 + \min\{\beta_3 \|v_t\|^2, \alpha_1\}\Big),$$
(29)

see App. B, Eqs. (45)–(46). Thus the method *automatically retreats from instability* as velocities spike near AEoS and prevents runaway steps. Because the gate is scalar, these guarantees are orthogonally invariant and do not require the adaptive preconditioner to commute with the Hessian. This mechanism aligns with the reduced ringing and sharpness observed empirically.

**Avoiding switched-anisotropy instabilities.** Allowing the step to vary per coordinate corresponds to a switched linear system with state matrix $A(D_t)$, where $D_t$ is a time-varying diagonal scaling. Even if each fixed $D$ yields a Schur-stable map (spectral radius $< 1$), products such as $A(D_2)A(D_1)$ can be *unstable* because the maps generally do not commute and the contraction directions rotate across steps. A scalar $\eta_t$ eliminates this failure mode: all directions are scaled identically, and the CQLF from Theorem 4.1 guarantees contraction under *arbitrary* scalar switching, again under the same Schur bound eq. 11 ( eq. 13–eq. 14).

## 4.4 Convergence Analysis

In this section, we provide a convergence analysis of VRAdam by extending the work of Défossez et al. (2022) on Adam. The main result is a probabilistic bound that VRAdam converges in expectation to a stationary point for a nonconvex (and therefore also a convex) objective function under a set of regularity assumptions and simplifications.

**Setup.** The objective function is given by $F(\theta) : \mathbb{R}^d \to \mathbb{R}$. $\nabla f$ are gradient estimates of $F$ such that $\mathbb{E}[\nabla f] = \nabla F$. The random variable $\tau_N$ describes an index with probabilities given by $\mathbb{P}[\tau = j] \propto 1 - \beta_1^{N-j}$ for $j \in \{0, 1, ..., N-1\}$. The constants $\alpha_{\min} = \alpha_0/(1 + \alpha_1) > 0$ and $\alpha_{\max} = \alpha_0 > 0$ correspond to the minimal and maximal possible learning rate at every step. $F$ is bounded from below, namely $\forall \theta, F(\theta) \ge F_*$. The gradient estimates are almost surely bounded in the $\ell_\infty$-norm, namely $\|\nabla f\|_\infty \le R - \sqrt{\epsilon}$. $\nabla F$ is L-Lipschitz-continuous in the $\ell_2$-norm, namely $\forall \theta, \phi, \|\nabla F(\theta) - \nabla F(\phi)\|_2 \le L \|\theta - \phi\|_2$. The weight decay parameter $\lambda = 0$ is set to zero. The bias correction for the velocity $\hat{v}_t \leftarrow v_t/(1 - \beta_1^t)$ is not included. The hyperparameter $\beta_1, \beta_2$ fulfill $0 < \beta_2 < 1, 0 \le \beta_1 < \beta_2$ and the number of iterates satisfies $N > \beta_1/(1 - \beta_1)$.

**Theorem 4.2** (Convergence of VRAdam). *Given the six assumptions above, the iterates $\theta_t$ of Algorithm 1 fulfill:*

$$\mathbb{E}\big[\|\nabla F(\theta_\tau)\|^2\big] \le 2R \frac{F(\theta_0) - F_*}{\alpha_{\min} \tilde{N}} \quad + E\Big(\frac{1}{N} \ln\big(1 + \frac{R^2}{(1-\beta_2)\epsilon}\big) - \frac{N}{N} \ln(\beta_2)\Big).$$
(30)

*with $\tilde{N} = N - \frac{\beta_1}{1-\beta_1}$, and*

$$E = \frac{\alpha_{\max}^2 dRL(1-\beta_1)}{\alpha_{\min}(1-\beta_1/\beta_2)(1-\beta_2)} + \frac{12\,\alpha_{\max} dR^2 \sqrt{1-\beta_1}}{\alpha_{\min}(1-\beta_1/\beta_2)^{3/2} \sqrt{1-\beta_2}} + \frac{2\,\alpha_{\max}^3 dL^2 \beta_1}{\alpha_{\min}(1-\beta_1/\beta_2)\,(1-\beta_2)^{3/2}}.$$
(31)

A proof can be found in the appendix C. The right-hand side of the bound converges to zero in the limit of $N \to \infty$ if $\alpha_{min}(N) \propto N^{-1/2}$, $\alpha_{max}(N) \propto N^{-1/2}$, and $1 - \beta_2(N) \propto N^{-1}$ to leading order. For this scaling of hyperparameters, VRAdam has the same convergence rate of $\mathcal{O}(\ln(N)/\sqrt{N})$ as Adam. While this convergence result provides a strong theoretical foundation, benchmarking is key to understanding performance under realistic conditions.

## 5 BENCHMARKS

We benchmark VRAdam against various optimizers on diverse datasets, tasks, and architectures. These benchmarks include image classification using a CNN on the CIFAR-10 dataset (Krizhevsky & Hinton, 2009), language modeling with a Transformer architecture (Vaswani et al., 2023) on the WikiText2 dataset (Merity et al., 2016), Generative Flow Networks (GFlowNets) (Bengio et al., 2021) on a grid world for sequence generation tasks (Chevalier-Boisvert et al., 2023), and training a large language model (LLM) such as the Generative Pretrained Transformer (GPT) (Brown et al., 2020). These benchmarks represent a broad variety of deep learning architectures and tasks, ranging from older techniques for image classification to much newer and actively emerging models and training techniques. We focus on the comparison with AdamW since its exceeding popularity and consistent performance but also report the performance of stochastic gradient descent (SGD) with momentum (Qian, 1999), root mean square propagation (RMSProp) (Ruder, 2017) and RAdam (Liu et al., 2020) for all tasks except for the LLM benchmark. In particular, we demonstrate the excellent convergence properties as well as performance improvements in most benchmarks for VRAdam (Table 1). Note that we run *Bayesian optimization for hyperparameter sweeps* for the image classification, and language modeling tasks, with several evaluations in close proximity to the found minimum, and the results are statistically significant. We report the optimal value from this optimization procedure, however, we include shaded error bars in the appendix. The Bayesian optimization for the hyperparameters was conducted with the objective to minimize validation loss, however, we also report test loss. Hyperparameters for the task involving GFlowNets were picked at random due to compute constraints. All details regarding model training, setup and datasets along with error envelopes using multiple runs are provided in Appendices E and G.

| Method | WikiText-2 Loss | | CIFAR-10 Loss | | GridWorld Flow Matching Loss | |
| --- | --- | --- | --- | --- | --- | --- |
| | Validation | Test | Validation | Test | Validation | Test |
| VRAdam | **5.99** | **6.00** | **0.476** | **0.469** | **1.25** | **1.33** |
| AdamW | 6.47 | 6.50 | 0.522 | 0.565 | 2.41 | 3.60 |
| RAdam | 7.511 | 7.554 | 2.300 | 4.005 | 1.407 | 2.290 |
| SGD+Nesterov | NaN | NaN | 0.625 | 0.620 | 2.71 | 2.61 |
| RMSProp | NaN | NaN | 0.801 | 0.813 | 25.0 | 25.0 |

Table 1: Comparison of optimizer performance across three tasks: language modeling on WikiText-2, image classification on CIFAR-10, and flow matching on GridWorld.

| Method | Training time (s) | Validation Loss |
| --- | --- | --- |
| AdamW | 48549.56 | 3.511 |
| VRAdam | 48522.40 | 3.447 |

Table 2: Comparison of training time and validation loss for GPT-2 training.

**LLM benchmark.** In order to demonstrate the effectiveness of VRAdam for training large networks, such as large language models (Brown et al., 2020), we present the validation loss results of VRAdam against that of AdamW and Lion when training a 124M parameter GPT-2 model on the FineWebEdu-10B dataset from scratch (Penedo et al., 2024). In Table 2, we report the validation loss after around 57000 steps or 2 epochs as well as the average training time per epoch on a H100 GPU using hyperparameters found in the Appendix.

| Setting | Model | Dataset | AdamW PPL | VRAdam PPL | Lion PPL |
| --- | --- | --- | --- | --- | --- |
| 4-bit QLoRA | LLaMA-2-7B | OASST2 | 3.84 | **3.55** | 3.56 |
| Full model | GPT-2 Large (774M) | GSM8K | 4.12 | **3.53** | 3.67 |

Table 3: Additional LLM benchmarks comparing AdamW, VRAdam, and Lion (Chen et al., 2023) in challenging fine-tuning regimes for OASST2 (instruction following) and GSM8K (math reasoning).

Beyond the 124M GPT-2 pretraining setup, we also evaluate VRAdam in two more challenging language-modeling regimes summarized in Table 3. First, we fine-tune a 7B LLaMA-2 model using 4-bit NF4 QLoRA on the OASST2 instruction-following dataset Touvron et al. (2023). This setting lies close to the adaptive edge-of-stability: gradients are both low-precision and strongly stochastic Hayou et al. (2024). We also perform full-parameter fine-tuning of GPT-2 Large (774M) on GSM8K. These experiments indicate that VRAdam scales favorably from medium-sized to larger LLMs, both in quantized low-rank adaptation and full-model fine-tuning settings. On GSM8K, fine-tuning GPT-2-Large with VRAdam improves exact-match accuracy from 35% (AdamW) to 42% under the same training budget. On OASST2, in a 4-bit QLoRA setup for LLaMA-2-7B, VRAdam improves an automatic instruction-following quality score from 72.3/100 to 78.5/100 compared to AdamW. This score combines lexical diversity, repetition, and response length over 50 held-out instructions.

## 6 RELATED WORK

**Follow-up work on Adam.** The Adam optimizer was introduced in 2015 by Kingma & Ba (2015) by combining the previously existing concepts of momentum and scaling a base LR for each parameter based on second-order moment estimates. The base learning rate, however, remains hard-coded (potentially chosen through a learning rate scheduler) throughout training. Since then, several modifications to Adam have been introduced, such as NAdam (Dozat, 2016), RAdam (Liu et al., 2020), Adabelief (Zhuang et al., 2020), and in particular AdamW (Loshchilov & Hutter, 2017), which reintroduces weight decay in its original intention. LARS (You et al., 2017) and LAMB (You et al., 2020) compute learning rates for layers individually. More recent optimization techniques include LION, an automatically discovered alternative to signed momentum (Chen et al., 2023), Sophia (Liu et al., 2023), which uses estimated diagonal entries of the Hessian as a precondition, sharpness-aware minimization methods (Foret et al., 2021), and a modified LR in Adam for reinforcement learning (Ellis et al., 2024).

**Understanding training dynamics, convergence analysis, and edge of stability.** Another line of work focuses on understanding the dynamics of the training of deep neural networks as well as derive convergence properties and guarantees for commonly used optimizers. Wang & Choromanska (2025) provides a recent survey over the later and Reddi et al. (2019) provide an explicit convex example for which Adam does not converge. A popularized framework for understanding training dynamics in the continuous training flow and infinite network width limit was introduced in Jacot et al. (2018), extended by finite width corrections in Huang & Yau (2020), and developed for graph neural networks in Du et al. (2019). Lastly, the role of the edge of stability regime offers an empirically view and was analyzed in Cohen et al. (2024); Arora et al. (2022); Cohen et al. (2022); Damian et al. (2022); Song & Yun (2023); Wang et al. (2022)

**Symplectic optimization.** This research area derives discrete-time optimization algorithms by discretizing continuous-time Hamiltonian or Lagrangian flows using symplectic integrators, which exactly preserve the underlying geometric (symplectic) structure of the dynamical system. This approach guarantees long-term stability, energy-preservation properties (or controlled energy dissipation in the dissipative case), and can provide valuable insights into existing optimizers. Notable work includes Betancourt et al. (2018); França et al. (2020); Maddison et al. (2018); Duruisseaux & Leok (2023); Yuan & Zhang (2023).

## 7 DISCUSSION

**Limitations.** The edge of stability regime remains not fully understood and questions regarding generalization capabilities remain. We also have constrained compute resources.

**Summary and future of work.** Motivated by physical perspectives for complex optimization scenarios and stability conditions along with the adaptive edge of stability, we developed a new optimizer VRAdam based on quartic kinetic energy terms. We analyzed its performance at the adaptive edge of stability and benchmark it against several optimizers in particular AdamW on image classification, language modeling, and a generative task using GFlowNets where we report improved performance and robustness to hyperparameters. We hope that this work leads to further development of interpretable optimizers using concepts from physics.

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

## A  BOUNDING NRQCD

By explicitly breaking certain symmetries—Lorentz invariance in NRQCD and time-translation in time crystals— higher-order kinetic terms paradoxically enhance stability through topological protection mechanisms and the generation of emergent length/time scales (Niemi, 2021; Guha & Ghose-Choudhury, 2019).

As a demonstration of this phenomenon, we can consider Nonrelativistic QCD (NRQCD), which is an effective field theory that expands full quantum chromodynamics in inverse powers of a heavy-quark mass $m$ (Assi et al., 2023). The bilinear heavy-quark Lagrangian $\mathcal{L}$ up to $O(1/m^3)$ reads,

$$\mathcal{L}_{\mathrm{NRQCD}} = \psi^\dagger \left( iD_0 - m + \frac{\mathbf{D}^2}{2m} + \frac{\mathbf{D}^4}{8m^3} + \cdots \right) \psi \tag{32}$$

The corresponding Hamiltonian density is derived via Legendre transformation:

$$\mathcal{H}_{\mathrm{NRQCD}} = \psi^\dagger \left( m - \frac{\mathbf{D}^2}{2m} - \frac{\mathbf{D}^4}{8m^3} - \cdots \right) \psi \tag{33}$$

For a rigorous analysis of the dispersion relation, we work in momentum space:

$$\psi(\vec{x}, t) = \int \frac{d^3p}{(2\pi)^3} e^{i\vec{p}\cdot\vec{x}} \tilde{\psi}(\vec{p}, t) \tag{34}$$

In momentum space, the operators transform as:

$$D_0 \to -iE \quad \text{(time evolution operator)} \tag{35}$$
$$\mathbf{D} \to i\vec{p} \quad \text{(spatial covariant derivative)} \tag{36}$$
$$\mathbf{D}^2 \to -p^2 \quad \text{(squared spatial derivative)} \tag{37}$$
$$\mathbf{D}^4 \to p^4 \quad \text{(quartic spatial derivative)} \tag{38}$$

The energy eigenvalue equation derived from the Hamiltonian is:

$$E\tilde{\psi}(\vec{p}) = \left( m + \frac{p^2}{2m} - \frac{p^4}{8m^3} + \mathcal{O}\left(\frac{1}{m^5}\right) \right) \tilde{\psi}(\vec{p}) \tag{39}$$

This gives the NRQCD dispersion relation to order $1/m^3$:

$$E_{\text{NRQCD}}(p) = m + \frac{p^2}{2m} - \frac{p^4}{8m^3} + \mathcal{O}\left(\frac{1}{m^5}\right) \tag{40}$$

## A.1 ANALYSIS OF BOUNDEDNESS

The exact relativistic energy-momentum relation:

$$E_{\text{rel}}(p) = \sqrt{m^2 + p^2} \tag{41}$$

Using Taylor series expansion for $\sqrt{1+x}$ where $x = p^2/m^2$:

$$\sqrt{1+x} = 1 + \frac{1}{2}x - \frac{1}{8}x^2 + \frac{1}{16}x^3 - \frac{5}{128}x^4 + \mathcal{O}(x^5) \tag{42}$$

Applying this to the relativistic energy:

$$E_{\text{rel}}(p) = m\left(1 + \frac{1}{2}\frac{p^2}{m^2} - \frac{1}{8}\frac{p^4}{m^4} + \mathcal{O}\left(\frac{p^6}{m^6}\right)\right) \tag{43}$$

Simplifying:

$$E_{\text{rel}}(p) = m + \frac{p^2}{2m} - \frac{p^4}{8m^3} + \mathcal{O}\left(\frac{p^6}{m^5}\right) \tag{44}$$

For the non-relativistic approximation without the quartic term:

$$\lim_{p \to \infty} \frac{E_{\text{NR}}(p)}{E_{\text{rel}}(p)} = \lim_{p \to \infty} \frac{m + \frac{p^2}{2m}}{\sqrt{m^2 + p^2}} \tag{45}$$

$$= \lim_{p \to \infty} \frac{m + \frac{p^2}{2m}}{p\sqrt{1 + \frac{m^2}{p^2}}} \tag{46}$$

$$= \infty \tag{47}$$

This shows that the non-relativistic approximation without the quartic term diverges from the true relativistic behavior.

The quartic term introduces a negative contribution to the energy that precisely cancels the fourth-order term in the relativistic expansion:

$$E_{\text{NRQCD}}(p) = m + \frac{p^2}{2m} - \frac{p^4}{8m^3} + \mathcal{O}\left(\frac{p^6}{m^5}\right) \tag{48}$$

Let's define a parameter $\lambda = p^2/m^2$ (proportional to $v^2$). For the NRQCD expansion to be valid, we require $\lambda \ll 1$.

The second derivative of the NRQCD energy with respect to $\lambda$ is:

$$\frac{d^2 E_{\text{NRQCD}}}{d\lambda^2} = -\frac{m}{4} + \mathcal{O}(\lambda) \tag{49}$$

the truncated dispersion is concave and has a finite maximum, so it does not increase unboundedly.

## B PHYSICS ANALYSIS

In this section, we provide a stability analysis from a physical perspective. Starting from the quartic Lagrangian in Eq. 3, we derive the one-dimensional dynamics and discretize them implicitly through the state-dependent step

$$\eta_t = \frac{\alpha_0}{1 + \min(\beta_3 |v_t|^2, \alpha_1)}, \qquad v_t = \beta_1 v_{t-1} + (1 - \beta_1)\nabla V(x_{t-1}), \qquad x_t = x_{t-1} - \eta_t v_t. \tag{50}$$

For a quadratic potential $V(x) = \frac{1}{2}x^\top H x$ with $0 \prec H \preceq LI$, we work in the eigenbasis $H = Q^\top \mathrm{diag}(h_i) Q$. Writing coordinates as $(x_i, v_i)$ and fixing $h \in (0, L]$, the one-dimensional update reads

$$v = \beta_1 u + (1 - \beta_1) h x, \qquad x^+ = x - \eta v, \qquad y := \eta h, \qquad \alpha := 1 - \beta_1, \tag{51}$$

where $u := v_{t-1}$ and $\eta \in [\eta_{\min}, \alpha_0]$ with $\eta_{\min} = \alpha_0/(1 + \alpha_1)$.

We introduce an energy that mirrors the quartic kinetic term and couples consistently to the curvature:

$$\mathcal{E}_h(x, v) = \frac{1}{2} h x^2 + \sigma x v + \frac{\gamma}{2h} v^2 + \frac{\delta}{4} v^4, \tag{52}$$

and set

$$\sigma = \zeta \frac{\beta_1}{2} \eta_{\min}, \qquad \gamma = \zeta \frac{\eta_{\min}}{1 - \beta_1}, \qquad \delta \geq \frac{\alpha_0^2}{\beta_3}, \qquad \zeta \in (0, 1). \tag{53}$$

A direct expansion of $\mathcal{E}_h(x^+, v) - \mathcal{E}_h(x, u)$ yields a quadratic form in $(x, u)$ plus the quartic difference $(v^4 - u^4)/4$. Because the coefficients of the quadratic form are convex polynomials in $y = \eta h$, the worst case over $y \in [\eta_{\min} h, \alpha_0 h]$ occurs at the endpoints. Evaluating there, and using the adaptive edge-of-stability bound for momentum,

$$\alpha_0 L < \frac{2(1 + \beta_1)}{1 - \beta_1}, \tag{54}$$

one obtains constants $c_x, c_u > 0$ (depending only on $\beta_1, \alpha_0, \eta_{\min}$) such that, uniformly for all $h \in [\mu, L]$ and $\eta \in [\eta_{\min}, \alpha_0]$,

$$\mathcal{E}_h(x^+, v) - \mathcal{E}_h(x, u) \leq -c_x h x^2 - c_u \frac{u^2}{h} + \frac{\delta}{4}(v^4 - u^4). \tag{55}$$

The gate implies $\eta \leq \alpha_0/(1 + \beta_3 v^2)$ and hence

$$\frac{1}{2}\eta^2 v^2 \leq \frac{\alpha_0^2}{8\beta_3}.$$

The choice $\delta \geq \alpha_0^2/\beta_3$ therefore ensures that whenever $|v| \geq 1/\sqrt{\beta_3}$ the quartic contribution dominates any potential increase from the discretization term and contributes strict dissipation; when $|v| \leq 1/\sqrt{\beta_3}$ the negative quadratic terms already control the step.

Summing over coordinates yields the global energy

$$\mathcal{E}(x, v) = \sum_{i=1}^{d} \mathcal{E}_{h_i}(x_i, v_i), \tag{56}$$

and constants $\kappa_1, \kappa_2, \kappa_4 > 0$ such that

$$\mathcal{E}(x_t, v_t) - \mathcal{E}(x_{t-1}, v_{t-1}) \leq -\kappa_1 x_{t-1}^\top H x_{t-1} - \kappa_2 v_{t-1}^\top H^{-1} v_{t-1} - \kappa_4 \sum_{i=1}^{d} \max\left\{0, v_{t,i}^4 - \frac{1}{\beta_3^2}\right\}. \tag{57}$$

This inequality formalizes the physical picture in Fig. 1. The first two terms express curvature-weighted exchange between potential and quadratic kinetic energy with net dissipation; the quartic term acts as a brake that activates in high-velocity regimes created near the adaptive edge of stability. The same mechanism explains the reduction in ringing and lower sharpness observed in Fig. 2: when $|v_t|$ grows, the gate reduces $\eta_t$ and the quartic channel increases dissipation, pushing the dynamics back to a low-velocity regime.

Two immediate consequences follow. First, the instantaneous adaptive stability threshold increases with the measured velocity:

$$L_{\mathrm{EoS}}(t) = \frac{2(1 + \beta_1)}{(1 - \beta_1)\eta_t} = \frac{2(1 + \beta_1)}{(1 - \beta_1)\alpha_0}\left(1 + \min(\beta_3 |v_t|^2, \alpha_1)\right), \tag{58}$$

so the method moves away from instability as oscillations grow.

Second, each parameter update is uniformly bounded:

$$|x_t - x_{t-1}| = \eta_t |v_t| = \frac{\alpha_0 |v_t|}{1 + \beta_3 |v_t|^2} \leq \frac{\alpha_0}{2\sqrt{\beta_3}}, \tag{59}$$

which prevents runaway steps and is not available to classical momentum. These properties are consistent with the design of Algorithm 1 and the empirical behavior reported in the analysis section.

## C  CONVERGENCE PROOF FOR VRADAM

The proof is based on the work of the proof in Défossez et al. (2022).

There, the authors derive in equation A.37 the bound

$$\underbrace{\frac{1}{2R} \sum_{n=1}^{N} \frac{\alpha_n}{\Omega_n} \sum_{k=0}^{n-1} \beta_1^k \, \mathbb{E}\big[\|G_{n-k}\|_2^2\big]}_{A} \leq F(x_0) - F_*$$

$$+ \underbrace{\alpha_N^2 L^2 \sum_{n=1}^{N} \mathbb{E}\big[\|u_n\|_2^2\big]}_{B}$$

$$+ \underbrace{\frac{\alpha_N^3 L^2}{4R\sqrt{1-\beta_1}} \sum_{n=1}^{N} \sum_{l=1}^{n-1} \mathbb{E}\big[\|u_{n-l}\|_2^2\big] \sum_{k=l}^{n-1} \frac{\beta_1^k}{\sqrt{k}}}_{C}$$

$$+ \underbrace{3\alpha_N R \sqrt{1-\beta_1} \sum_{n=1}^{N} \sum_{k=0}^{n-1} \left(\frac{\beta_1}{\beta_2}\right)^k \sqrt{k+1} \, \mathbb{E}\big[\|U_{n-k}\|_2^2\big]}_{D}.$$

By bounding $\alpha_n$ with $\alpha_{\min}$ in expression $A$, and $\alpha_N$ with $\alpha_{\max}$ in expressions $B, C, D$ we obtain the final result. Although not included in the proof, the numerical impact of the velocity bias correction becomes negligible after the first few update steps and asymptotic behavior remains unchanged (Défossez et al., 2022). It is well established that weight decay contributes to improved robustness and generalization in practice and is therefore included here, despite a challenging theoretical analysis.

## D  ABLATION ON KINETIC ENERGY POTENTIAL

Table 4 displays the test loss for higher powers $k$ in the learning rate $\alpha_0/(1 + \min(\beta_3||v_t||^k, \alpha_1))$, which correspond to fifth- to eighth-order terms in the kinetic energy. Training was performed on CIFAR-10 for 10 Epochs with $\alpha_0 = 0.005$.

Table 4: Test Loss on CIFAR-10 for different kinetic energies

| Power | Test Loss |
|---|---|
| 2 | **0.932** |
| 3 | 1.155 |
| 4 | 0.974 |
| 5 | 1.004 |
| 6 | 1.036 |

## E  HYPERPARAMETERS, DATASETS AND MODEL ARCHITECTURES

We recommend $\beta_3 = 1.0$ and $\alpha_1 = 10$ as the default setting for VRAdam.

## E.1 IMAGE CLASSIFICATION

This section details the comprehensive sweep for the CNN, on the CIFAR-10 dataset.

- Model: Convolutional Neural Network
- Dataset: CIFAR-10
- Hyperparameter sweep method: Bayesian optimization
- Optimization metric: validation loss

Table 5: Dataset Splits during CNN Sweep

| Dataset Split | Configuration Description |
|---|---|
| Training | 80% of the original CIFAR-10 training set (50,000 images). |
| Validation | 20% of the original CIFAR-10 training set. |
| Test | Full CIFAR-10 test set (10,000 images). |

Table 6: Fixed Hyperparameters during CNN Comprehensive Sweep

| Parameter | Value |
|---|---|
| Model Architecture | Convolutional Neural Network |
| Dataset | CIFAR-10 |
| Epochs | 100 |
| Batch Size | 1024 |
| Scheduler Type (AdamW) | WarmupCosineAnnealing |
| Warmup Epochs | 5 |
| Warmup Factor | 0.1 |
| Scheduler $\eta$ min | $1 \times 10^{-5}$ |
| VRAdam $\beta_1$ | 0.9 |
| VRAdam $\beta_2$ | 0.999 |
| VRAdam power | 2 |
| VRAdam weight decay | $1 \times 10^{-5}$ |
| VRAdam $\epsilon$ | $1 \times 10^{-8}$ |
| AdamW $\beta_1$ | 0.9 |
| AdamW $\beta_2$ | 0.999 |
| AdamW weight decay | $1 \times 10^{-5}$ |
| SGD momentum | 0.9 |
| SGD nesterov | True |
| SGD weight decay | $1 \times 10^{-5}$ |
| RMSProp $\alpha$ | 0.99 |
| RMSProp | $1 \times 10^{-8}$ |
| RMSProp weight decay | $1 \times 10^{-5}$ |

Table 7: Swept Hyperparameters during CNN Comprehensive Sweep

| Optimizer | Parameter | Sweep Configuration | Optimal Parameter |
|---|---|---|---|
| VRAdam | $\alpha_0$ | Log-uniform, Min: $1 \times 10^{-4}$, Max: 0.1 | 0.0846 |
| VRAdam | $\beta_3$ | Uniform, Min: 0.1, Max: 1.5 | 1.015 |
| VRAdam | $\alpha_1$ | Integer Uniform, Min: 3, Max: 30 | 29 |
| AdamW | $\eta$ | Log-uniform, Min: $1 \times 10^{-5}$, Max: $1 \times 10^{-1}$ | 0.0625 |
| SGD | $\eta$ | Log-uniform, Min: $1 \times 10^{-5}$, Max: $1 \times 10^{-1}$ | 0.00784 |
| RMSProp | $\eta$ | Log-uniform, Min: $1 \times 10^{-5}$, Max: 0.1 | 1.78e-4 |

## E.2 LANGUAGE MODELING

This section details the comprehensive sweep for the Transformer, on the WikiText-2 dataset.

- Model: Transformer
- Dataset: Wikitext-2
- Hyperparameter sweep method: Bayesian optimization
- Optimization metric: validation loss

Table 8: Dataset Splits during Transformer Comprehensive Sweep

| Dataset Split | Configuration Description |
|---|---|
| Training | Full WikiText-2 predefined training set. |
| Validation | Full WikiText-2 predefined validation set. |
| Test | Full WikiText-2 predefined test set. |

Table 9: Fixed Hyperparameters during Transformer Comprehensive Sweep

| Parameter | Value |
|---|---|
| Model Architecture | TransformerModel |
| Epochs | 100 |
| Batch Size | 32 |
| Seed | 0 |
| Scheduler Type | WarmupCosineAnnealing |
| Warmup Epochs | 5 |
| Warmup Factor | 0.1 |
| Scheduler $\eta$ | $1 \times 10^{-5}$ |
| Model sequence length | 64 |
| Model embed dimension | 128 |
| Model hidden dimension | 256 |
| VRAdam $\beta_1$ | 0.9 |
| VRAdam $\beta_2$ | 0.999 |
| VRAdam power | 2 |
| VRAdam weight decay | $1 \times 10^{-5}$ |
| VRAdam $\epsilon$ | $1 \times 10^{-8}$ |
| AdamW $\beta_1$ | 0.9 |
| AdamW $\beta_2$ | 0.999 |
| AdamW weight decay | $1 \times 10^{-5}$ |
| SGD sgd momentum | 0.9 |
| SGD sgd nesterov | True |
| RMSProp $\alpha$ | 0.1 |
| RMSProp $\epsilon$ | $1 \times 10^{-8}$ |

Table 10: Swept Hyperparameters during Transformer Comprehensive Sweep

| Optimizer | Parameter | Sweep Configuration | Optimal Parameter |
|---|---|---|---|
| VRAdam | $\alpha_0$ | Log-uniform, Min: $1 \times 10^{-5}$, Max: 0.1 | 1.55e-05 |
| VRAdam | $\beta_3$ | Uniform, Min: 0.1, Max: 5.0 | 3.35 |
| VRAdam | normgrad | Values: [True, False] | False |
| VRAdam | $\alpha_1$ | Integer Uniform, Min: 5, Max: 30 | 7 |
| Adam | $\eta$ | Log-uniform, Min: $1 \times 10^{-5}$, Max: 0.1 | 1.661e-05 |
| SGD | $\eta$ | Log-uniform, Min: $1 \times 10^{-5}$, Max: $1 \times 10^{-1}$ | - |
| RMSProp | $\eta$ | Log-uniform, Min: $1 \times 10^{-5}$, Max: 0.1 | - |

### E.3 GENERATIVE MODELING WITH GFLOWNETS

Here we include the hyperparameters used for reporting the performance of the optimizers for the GFlowNet.

Table 11: Hyperparameters GFlowNets

| Optimizer | Parameter | Value |
|---|---|---|
| VRAdam | $\alpha_0$ | 0.01 |
| VRAdam | $\beta_3$ | 1 |
| VRAdam | normgrad | False |
| VRAdam | $\alpha_1$ | 19 |
| VRAdam | weight decay | $1 \times 10^{-5}$ |
| AdamW | weight decay | $1 \times 10^{-5}$ |
| AdamW | $\eta$ | 0.01 |
| SGD | $\eta$ | 0.01 |
| RMSProp | $\eta$ | 0.01 |

## E.4 EDGE OF STABILITY ANALYSIS

Table 12: Hyperparameters edge of stability analysis

| Parameter | Value |
|---|---|
| Model Architecture | ResNet-32 |
| Max iterations | 20000 |
| Batch Size | 1000 |
| Seed | 0 |
| Loss criterion | Mean squared error |
| VRAdam $\alpha_0$ | 0.002 |
| VRAdam $\beta_1$ | 0.9 |
| VRAdam $\beta_2$ | 0.999 |
| VRAdam $\beta_3$ | 1 |
| VRAdam power | 2 |
| VRAdam normgrad | False |
| VRAdam $\alpha_1$ | 19 |
| VRAdam $\epsilon$ | $1 \times 10^{-7}$ |
| Adam $\beta_1$ | 0.9 |
| Adam $\beta_2$ | 0.999 |
| Adam $\epsilon$ | $1 \times 10^{-7}$ |

## F COMPUTE RESOURCES

All experiments were run on Lambda cloud instances or the Google cloud platform (GCP). Experiments were conducted either using a NVIDIA L4 GPU with 24 GB of GPU memory and 31 GB of system memory or larger experiments were performed on a NVIDIA A10 with 24 GB of GPU memory and 200 GB of system memory. The GPT benchmark was run on an NVIDIA H100 GPU.

## G LOSS CURVES WITH ERROR ENVELOPES

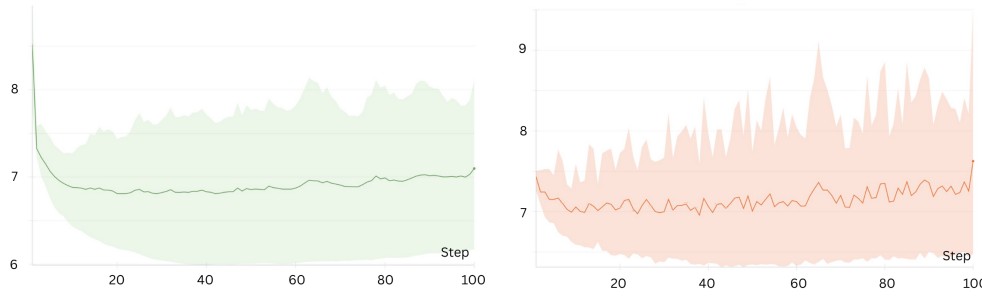

Figure 3: Train (left) and validation (right) loss curves with error envelopes calculated using different run values for language modeling using AdamW.

Train and validation loss curves calculated using different run values for language modeling using SGD Nesterov with momentum all generate NaN values. Visualization is not included.

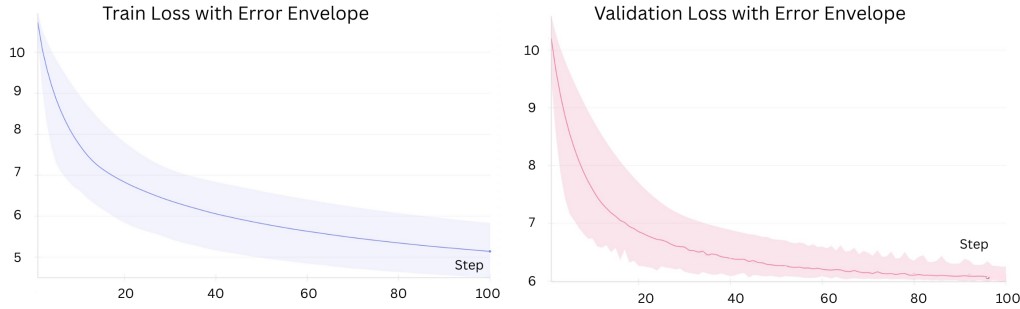

Figure 4: Train (left) and validation (right) loss curves with error envelopes calculated using different run values for language modeling using VRAdam.

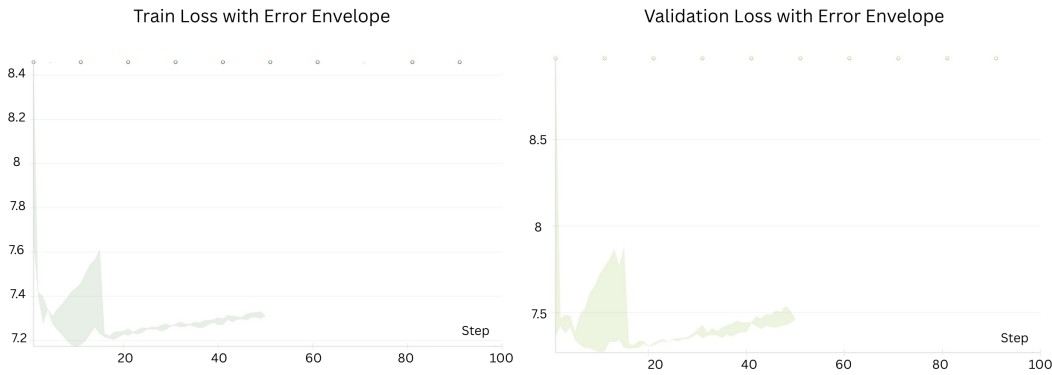

Figure 5: Train (left) and validation (right) loss curves with error envelopes calculated using different run values for language modeling using RMSProp. The dots on the top indicate NaN values.

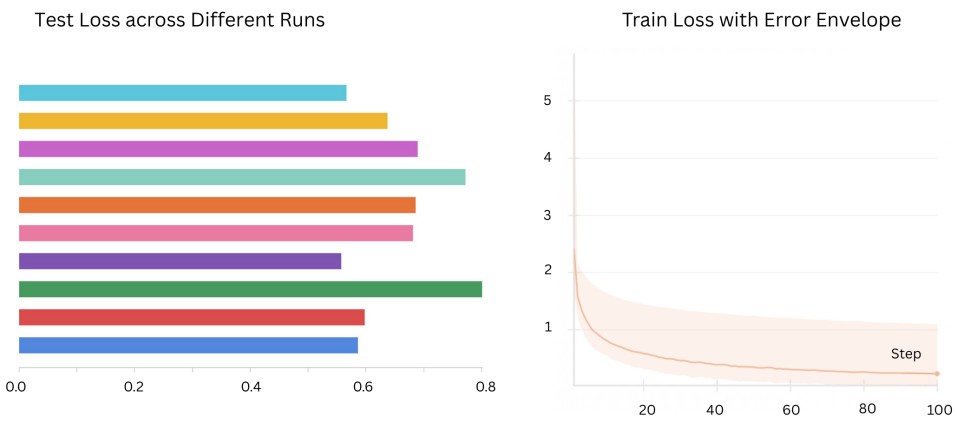

Figure 6: Train (right) curves with error envelopes calculated using different run values for image classification using VRAdam. Test loss values (left) over different runs.

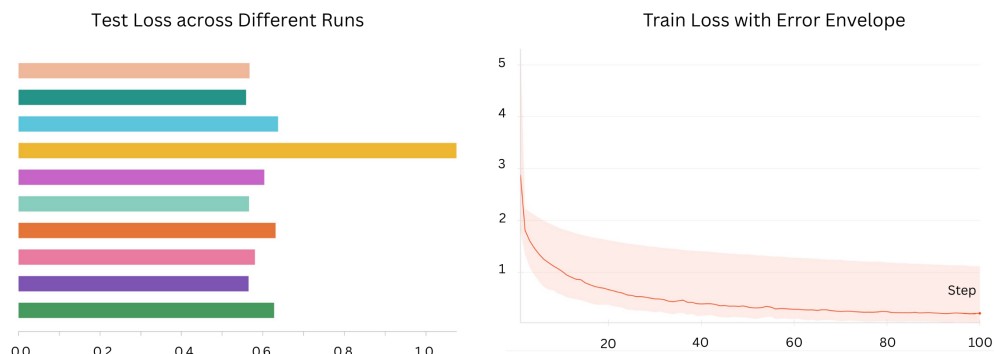

Figure 7: Train (right) curves with error envelopes calculated using different run values for image classification using AdamW. Test loss values (left) over different runs.

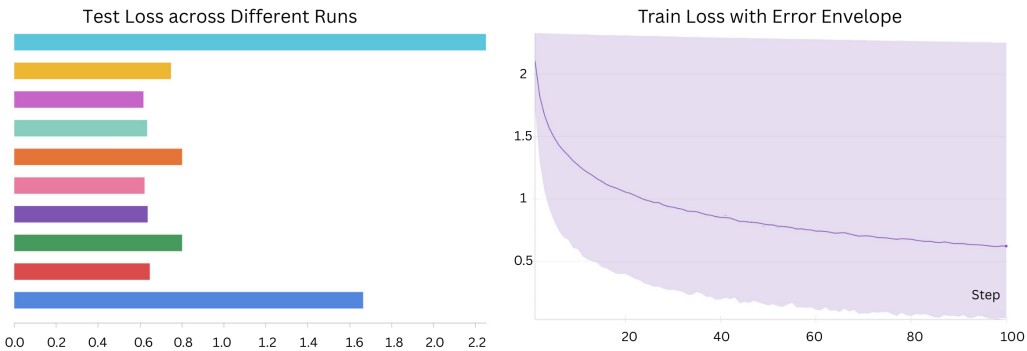

Figure 8: Train (right) curves with error envelopes calculated using different run values for image classification using SGD Nesterov with Momentum. Test loss values (left) over different runs.

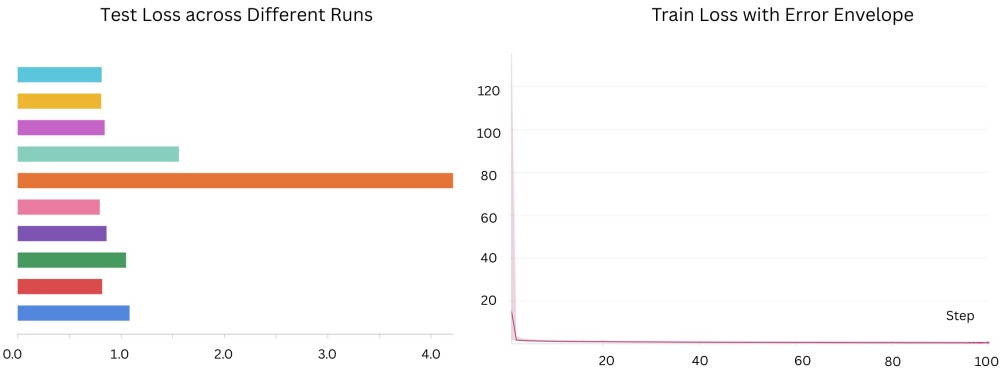

Figure 9: Train (right) curves with error envelopes calculated using different run values for image classification using RMSProp. Test loss values (left) over different runs.

## H  BACKGROUND ON PHASE SPACE, LAGRANGIANS, AND HAMILTONIANS

### H.1  PHASE SPACE

In classical mechanics, the state of a dynamical system is completely described by a point in a high-dimensional abstract space called **phase space** Kawai et al. (2007). For a system with $N$ degrees of freedom (e.g., the positions of multiple particles), the phase space is a $2N$-dimensional space. Its axes correspond to the generalized coordinates $x_i$ (representing positions) and their corresponding generalized momenta $p_i$. Each point $(x, p)$ in phase space represents a unique, instantaneous state of the system. The evolution of the system over time is then visualized as a trajectory traced out by this point moving through phase space.

### H.2  LAGRANGIAN FORMALISM

The Lagrangian formulation of classical mechanics describes the dynamics of a system using generalized coordinates and their time derivatives (velocities) Iro & Anderson (2003). The central quantity in this formalism is the **Lagrangian**, $\mathcal{L}$, which is a function of the system's coordinates and velocities. It is typically defined as the difference between the system's kinetic energy, $T$, and its potential energy, $V$:

$$\mathcal{L}(x, v) = T(v) - V(x) \tag{60}$$

where $v = \dot{x}$ represents the velocity. The system's path through its configuration space is determined by the **Principle of Least Action**. This principle states that the actual trajectory taken by the system between a starting time $t_1$ and an ending time $t_2$ is the one that minimizes the action integral, $S$:

$$S = \int_{t_1}^{t_2} \mathcal{L}(x, \dot{x}, t)\, dt \tag{61}$$

Applying the calculus of variations to find the path that minimizes this action integral yields the fundamental **Euler-Lagrange equations of motion**:

$$\frac{d}{dt}\frac{\partial \mathcal{L}}{\partial v} - \frac{\partial \mathcal{L}}{\partial x} = 0 \tag{62}$$

Evaluating derivatives requires treating $x$ and $\dot{x} = v$ as independent variables. This set of second-order differential equations fully defines the system's trajectory. In the context of deep learning optimization, the potential energy $V(x)$ is analogous to the loss function $L_{\text{loss}}(\theta)$, with the model parameters $\theta$ serving as the coordinates $x$. Optimizing the loss function then corresponds to solving the system of differential equations with a numerical integrator.

### H.3  HAMILTONIAN FORMALISM

The Hamiltonian formalism offers an alternative, often more powerful, description of system dynamics that is set within phase space, using coordinates $x$ and momenta $p$ as its fundamental variables. The transition from the Lagrangian to the Hamiltonian framework is achieved via a mathematical procedure known as a **Legendre transformation** Helliwell & Sahakian (2020). First, the **generalized (or canonical) momentum** $p$ is defined as the partial derivative of the Lagrangian with respect to the velocity:

$$p = \frac{\partial \mathcal{L}}{\partial v} \tag{63}$$

The **Hamiltonian**, $\mathcal{H}$, is then defined as:

$$\mathcal{H}(x, p) = \sum_i p_i v_i - \mathcal{L}(x, v) \tag{64}$$

For many standard physical systems where kinetic energy is a quadratic function of velocity and potential energy is a function of position only, the Hamiltonian is equivalent to the total energy of the system, $\mathcal{H} = T + V$. The dynamics are then described by a pair of first-order differential equations known as **Hamilton's equations**:

$$\dot{x} = \frac{\partial \mathcal{H}}{\partial p} \tag{65}$$

$$\dot{p} = -\frac{\partial \mathcal{H}}{\partial x} \tag{66}$$

These equations provide an elegant description of the flow of system states through phase space. A key feature of this formalism is that it naturally describes systems that conserve energy (if $\mathcal{H}$ is time-independent) and preserve the volume of phase space. This makes the Hamiltonian framework exceptionally well-suited for analyzing the long-term stability of dynamical systems and serves as the foundation for **symplectic optimization** methods.

## I    FURTHER DISCUSSION

Recent work by Defazio et al. (2024) introduces *Schedule-Free* optimizers (e.g., Schedule-Free SGD and AdamW), which remove the need for explicit learning rate schedules by utilizing a specific weighting of the iterate sequence. While both VRAdam and Schedule-Free methods aim to simplify hyperparameter tuning by modifying how the step size and updates are handled, they operate through fundamentally different mechanisms and target distinct dynamical regimes.

**Mechanism: Averaging vs. Feedback Control.** The core innovation of Schedule-Free methods is an interpolation between primal and Polyak averaging, where the effective learning rate is determined by a deterministic, time-dependent weighting scheme derived from online-to-batch conversion guarantees. The effective step size in these methods is a function of the iteration index $t$, simulating a decay schedule without a fixed horizon.

In contrast, VRAdam employs a *state-dependent* feedback mechanism. Our gating term, $\eta_t = \alpha_0/(1 + \beta_3 \|v_t\|^2)$, is not a explicit function of time, but strictly a function of the global momentum buffer norm. Consequently, VRAdam reacts dynamically to the optimization trajectory: it actively damps updates during periods of high oscillatory kinetic energy (common in the early training phase or high-curvature regions) and relaxes the gate when the trajectory stabilizes. This creates a closed-loop feedback control system rather than a pre-computed averaging schedule.

**Theoretical Focus.** The theoretical underpinning of Defazio et al. (2024) focuses on achieving worst-case optimal convergence rates for convex Lipschitz problems. Our analysis focuses on the *Adaptive Edge of Stability* (AEoS) (Cohen et al., 2024). We derive our update rule from a quartic Lagrangian to ensure global exponential stability by enforcing a uniform bound on the parameter update norm, $\|\theta_t - \theta_{t-1}\|$, effectively raising the stability threshold in response to gradient bursts.

Finally, we note that these approaches are mechanistically orthogonal. The velocity-regularized gate of VRAdam operates on the preconditioned gradient and could, in principle, be combined with the iterate averaging schemes of Schedule-Free methods, though we leave such hybrid explorations to future work.

