# OpenReview forum: "A Physics-Inspired Optimizer: Velocity Regularized Adam"
_ICLR.cc/2026/Conference — ICLR 2026 Poster_

### Official Review · Reviewer_uYkK · 2025-10-21

**Soundness:** 3
**Presentation:** 2
**Contribution:** 3
**Rating:** 8
**Confidence:** 3

**Summary:**

This paper proposes Velocity-Regularized Adam (VRAdam), a new optimizer for deep neural network training that introduces a physics-inspired regularization mechanism. Based on the classical Adam optimizer, VRAdam incorporates a quartic kinetic energy term to dynamically regulate the effective learning rate based on the velocity (momentum) of parameter updates. The resulting learning rate shrinks automatically in high-velocity regimes, reducing oscillations and improving convergence stability. The paper provides a theoretical framework, proving uniform exponential stability via Lyapunov analysis for stochastic non-convex objectives. Extensive experiments on CIFAR-10, WikiText-2, GridWorld, and GPT-2 fine-tuning show that VRAdam achieves faster convergence, smoother training curves, and better generalization compared to AdamW and other optimizers.

**Strengths:**

1. The introduction of a quartic kinetic energy term as a stabilizing mechanism is a fresh and interesting physics-based perspective on optimizer design.

2. The analogy between optimization trajectories and particle dynamics adds an intuitive understanding of how the method moves away from instability near the edge of stability.

3. The paper rigorously proves global uniform exponential stability and convergence under mild conditions, supported by clear mathematical derivations.

4. VRAdam consistently outperforms AdamW, RAdam, RMSProp, and SGD across diverse tasks (CNNs, Transformers, GFlowNets, and LLMs).

**Weaknesses:**

1. The presentation is not friendly to readers not familiar with optimization techniques using Langevin dynamics.

2. Lots of hyperparameters are used. Although $\beta_3$ is claimed to be robust, the practical sensitivity of VRAdam to its hyperparameters $like (\alpha_0,\alpha_1,\beta_3)$ is not fully explored. Are they same with Adam or will be influenced by $\beta_3$ ?

3. While AdamW(2017) is a strong baseline, the study omits comparisons with newer optimizers (e.g., LION, AdaHessian), which are relevant for modern deep learning tasks.

4. The heavy use of physical analogies (NRQCD, Lagrangians) brings difficulties to understand the motivation and improvement of the algorithm for readers unfamiliar with physics.

5. The experiments only report the validation and test loss, the improvement against AdamW seems modest, whether VRAdam improves the task’s accuracy is not reported. Besides, no clear ablation on the contribution of the quartic term versus standard momentum damping—this would help isolate the true effect of velocity regularization.

**Questions:**

1. See weakness.

2. The author choose the quartic kinetic term NRQCD system as T(v), is there any other choice? Can you report the ablation study since it deserves to be the key insights of the improvement against AdamW.

3. How sensitive is VRAdam to the choice of the velocity penalizer β₃? Does it generalize well across tasks without tuning?

4. Does the quartic kinetic term introduce any bias that could affect convergence to flatter minima or generalization in practice? Can it be integrated with other techiques to improve generalization like in ‘Improving Generalization of Deep Neural Networks by Optimum Shifting , AAAI25’?

---

> ### Author Response · Authors · 2025-11-21
> **Response to concerns of reviewer uYkK**
>
> We thank the reviewer for their valuable feedback and their interest in our work. We apppreciate that the reviewer acknowledges the contributions of this work, particularly in introducing an intuitive and physics based optimizer.
>
> *The presentation is not friendly to readers not familiar with optimization techniques using Langevin dynamics. /W5 The heavy use of physical analogies (NRQCD, Lagrangians) brings difficulties to understand the motivation and improvement of the algorithm for readers unfamiliar with physics.*
>
> - We apologize that the physics analogies were difficult to follow. We have added a Background Section Appendix H ("Background on Phase Space, Lagrangians, and Hamiltonians") to the appendix. This section builds the intuition from first principles to follow along in the paper. Langevin dynamics are only used when referring to Neural Tanget Kernels and not crucial for the overall paper.
>
>
> *Lots of hyperparameters are used. Although β₃ is claimed to be robust, the practical sensitivity of VRAdam to its hyperparameters *like* (α₀, α₁, β₃) is not fully explored. Are they same with Adam or will be influenced by β₃? /Q3 How sensitive is VRAdam to the choice of the velocity penalizer $\beta_3$? Does it generalize well across tasks without tuning?*
>
> - We have included practical reccomendations for $\beta_3$ after extensive hyperparameter sweeps in Appendix E that we would like to highlight here. We observe that our hyperparameters provide stable training for a large range of tasks. As mentioned in the overall comment, we report that performance is much more sensitive to the learning rate of AdamW than to $\alpha_0$ in VRAdam, with even lower sensitivity to $\alpha_1$ and $\beta_3$, confirming the latter’s robustness.  Another theoretical reason for choosing $\beta_3$, follows from Appendix B. From the physics-inpsired Lyapunov construction, we can see that allowing  $\beta_3$ to be very small forces $\delta$  to be very large, this could cause the Lyapunov function to become very steep in $v$.
>
> *The experiments only report the validation and test loss; the improvement against AdamW seems modest. Whether VRAdam improves the task’s accuracy is not reported.*
>
> - For two further challenging language modelling fine-tuning task we also report perplexity with significant improvements.
>
>
> | Setting | Model | Dataset/Task |AdamW PPL| VRAdam PPL|
> | -------- | -------- | -------- |--------|---------|
> | 4-bit QLoRA     | LLaMA-2-7B     | OASST2 (Instruction Following)     |3.84    | **3.55**|
> |Full model |GPT-2 Large (774M)|GSM8k (Reasoning)|4.12|**3.53**|
>
>
>
>
>
> *The author chose the quartic kinetic term NRQCD system as T(v). Is there any other choice? Can you report the ablation study since it deserves to be the key insights of the improvement against AdamW?*
>
>
>  - While there are of course more choices the quartic kinetic energy is known to generate meta-stable states in various physical settings as seen from lattic QCD to time crystals (Shapere *et al.*, classical time crystals). We added an ablation over higher‑order kinetic terms (powers 2–6 in $||v||$), showing that the quartic‑based choice used in VRAdam is competitive and that very high powers do not systematically improve performance, supporting our physics‑motivated design.
>
>
> | **Power** | **Test Loss** |
> |-----------|----------------|
> | 2         | **0.932**      |
> | 3         | 1.155          |
> | 4         | 0.974          |
> | 5         | 1.004          |
> | 6         | 1.036          |
>
>
> *Does the quartic kinetic term introduce any bias that could affect convergence to flatter minima or generalization in practice? Can it be integrated with other techniques to improve generalization like in *“Improving Generalization of Deep Neural Networks by Optimum Shifting,” AAAI25*? *
>
> - Yes, we empirically observe lower test losses corresponding to better generalization. Since the optimal shifting method is applied after training, it can be integrated into VRAdam without any modifications.
>
> We hope these changes further strength the paper.

---

> > ### Comment · Reviewer_uYkK · 2025-11-24
> >
> > Thanks for the response. Most of my concerns have been solved. For W3, can the author share more insights on the comparison to some modern deep learning optimizers?

---

> > > ### Author Response · Authors · 2025-11-26
> > >
> > > We thank the reviewer for great feedback! We compare the performance of VRAdam against more optimizers from the literature, such as Lion, on the more recently included tasks of the paper.
> > >
> > > | Setting     | Model              | Dataset/Task                   | AdamW PPL | VRAdam PPL | Lion PPL |
> > > | ----------- | ------------------ | ------------------------------ | --------- | ---------- | -------- |
> > > | 4-bit QLoRA | LLaMA-2-7B         | OASST2 (Instruction Following) | 3.84      | **3.55**   | 3.56     |
> > > | Full model  | GPT-2 Large (774M) | GSM8k (Reasoning)              | 4.12      | **3.53**   | 3.67     |
> > >
> > > We did not include AdaHessian, due to the computational overhead associated with computing the diagonal Hessian operator using the Hutchinson method. VRAdam performs well in the edge of stability without any explicit preconditioning.

---

> > > > ### Comment · Reviewer_uYkK · 2025-11-28
> > > > **Thank you**
> > > >
> > > > Thanks for the reply. My concerns have been resolved. I keep my positive score and tend to acceptence of this paper.

---

### Official Review · Reviewer_JcRv · 2025-10-21

**Soundness:** 3
**Presentation:** 2
**Contribution:** 3
**Rating:** 6
**Confidence:** 3

**Summary:**

This paper proposes a physics-inspired optimizer, VRAdam, designed to address the oscillation and convergence slowdown problems that occur when adaptive optimizers (such as Adam) train near the stability boundary. Its core idea originates from kinetic models in classical physical systems. By introducing a velocity regularization term, the method imposes a higher-order penalty on the momentum (velocity) during optimization, thereby dynamically adjusting the learning rate and suppressing instability.

Contributions
1. Proposes a physics-inspired optimizer, VRAdam, which introduces a velocity regularization term into Adam to dynamically adjust the learning rate and suppress training oscillations.

2. Theoretically proves the stability of a simplified model and establishes a convergence bound under non-convex settings.

3. Demonstrates the effectiveness of the method on image classification, language modeling, generative flow networks, and GPT-2 training tasks.

**Strengths:**

1. The paper is not a minor modification of Adam but introduces an intuition derived from a non-standard Lagrangian containing a $v^4$ term. This physics-to-algorithm mapping is uncommon in existing literature and demonstrates novelty.

2. Theoretically, the paper provides a proof of uniform exponential stability for VRMomentum under quadratic objectives (Theorem 4.1) and establishes a convergence rate bound under stochastic non-convex settings comparable to that of Adam (Theorem 4.2).

3. The experimental tasks cover a wide range of applications.

4. If VRAdam proves to be stable and effective under broader settings (including larger-scale models and more repeated runs), it could be a strong improvement over existing optimizers.

**Weaknesses:**

1. The theoretical analysis is based on a simplified model, while the actual implementation includes all modules. This may lead to inconsistencies between theory and practice.

2. The tables lack explanations regarding reproducibility and do not include error bars, raising questions about the reliability of the results.

3. The authors are advised to discuss the robustness of the results to hyperparameters, as all experiments use a single configuration. In all experiments, β₃ is fixed to 1, but the rationale behind this choice is not provided.

4. The baseline methods are relatively outdated and insufficient in number. Although the related work section mentions more recent optimization techniques, these newer methods are not included in the experiments.

5. The authors only compared the training time between AdamW and VRAdam on GPT-2, and the results did not show a clear time advantage.

**Questions:**

1. The visualization of results could be improved. Why is Figure 5 in the appendix blank, and why are some figure fonts too small?

2. The theoretical proofs are based on a simplified model, while the actual implementation includes all modules. The authors are encouraged to explicitly discuss how this simplification affects the stability and convergence conclusions.

3. The reproducibility and statistical significance of the results are difficult to assess.

4. Did the authors conduct any study or experiments regarding the choice of $\beta_3$?

5. Have the authors attempted experiments on more complex tasks?

6. Does VRAdam provide any advantage in training time, and does it introduce additional computational overhead?

---

> ### Author Response · Authors · 2025-11-21
> **Responses to concerns of reviewer JcRv**
>
> We thank the reviewer for their constructive feedback and for highlighting several important aspects of our theoretical and empirical analysis. We have revised the manuscript to address these concerns and clarify the points raised.
>
>
> *The theoretical analysis is based on a simplified model.../Q2. The theoretical proofs are based on a simplified model, while the actual implementation includes all modules....*
>
> - We have significantly strengthened the analysis in Section 4.2 in the revised version. We explicitly added a "Nonconvex stability via analytical shift" analysis. We replace the global convexity assumption with a trajectory-level curvature bound, $-m I<\nabla^2 f\left(\theta_t\right)<L I$,  and include decoupled weight decay in this setting $\lambda>m$. Under these conditions, the condition $\alpha_0((1-\beta) L+(1+ \beta) \lambda)<2(1+\beta)$ is sufficient for every eigen-direction block to be Schur-stable uniformly over $h \in [-m, L]$. Since the momentum update encompasses the majority of our understanding around the edge-of-stability, this analysis has been further solidified. For the convergence analysis, we added a discussion to the the appendix. The  numerical impact of the velocity bias correction becomes negligible after the first few update steps and asymptotic behavior remains unchanged (Défossez et al. 2022). It is well established that weight decay contributes to improved robustness and generalization in practice and is therefore included here, despite a challenging theoretical analysis.
>
> *The tables lack explanations regarding reproducibility and do not include error bars.../Q3. The reproducibility and statistical significance of the results are difficult to assess*
>
> - We have increased the font size and clarity of the figures and now show clear error envelopes representing standard deviations across multiple runs. This confirms that the improvements shown by VRAdam are statistically significant and not artifacts.
>
>
> *The authors are advised to discuss the robustness of the results to hyperparameters..../Q4. Did the authors conduct any study or experiments regarding the choice of $\beta_3$*
>
> - We had already included practical reccomendations for $\beta_3$ after extensive hyperparameter sweeps in Appendix D that we would like to highlight here. We observe that our hyperparameters provide stable training for a large range of tasks. As mentioned in the overall comment, we report that performance is much more sensitive to the learning rate of AdamW than to $\alpha_0$ in VRAdam, with even lower sensitivity to $\alpha_1$ and $\beta_3$, confirming the latter’s robustness.  Another theoretical reason for choosing $\beta_3$, follows from Appendix B. From the physics-inpsired Lyapunov construction, we can see that allowing  $\beta_3$ to be very small forces $\delta$  to be very large, this could cause the Lyapunov function becomes very steep in $v$.
>
> *The authors only compared the training time between AdamW and VRAdam on GPT-2, and the results did not show a clear time advantage. Does VRAdam provide any advantage in training time, and does it introduce additional computational overhead?*
>
> - When benchmarking the time until a certain loss target is reached, VRAdam provides a significant advantage (e.g. see Figure 2). Most of the benchmarks, however, are designed to compare the loss after a fixed number of epochs for all optimisers. Therefore, when training is performed until a target is reached, VRAdam provides an advantage. Theoretically, the overhead of computation boils down to evaluating a single norm of the velocity vector once for each optimizer step. This is a small term (linear in the number of parameters) negligible during training. There are no additional memory requirements. We confirm this experimentaly by providing similar training times for VRAdam and AdamW for GPT-2.
>
> *Have the authors attempted experiments on more complex tasks?*
>
> We also evaluate VRAdam in two more challenging language-modeling regimes. First, we fine-tune a 7B LLaMA-2 model using 4-bit QLoRA on the OASST2 instruction-following dataset. This setting lies close to the adaptive edge-of-stability: gradients are both low-precision and strongly stochastic. We also perform full-parameter fine-tuning of GPT-2 Large (774M) on GSM8K. These experiments indicate that VRAdam scales favorably from medium-sized to larger LLMs, both in quantized low-rank adaptation and full-model fine-tuning settings.
>
> | Setting | Model | Dataset/Task |AdamW PPL| VRAdam PPL|
> | -------- | -------- | -------- |--------|---------|
> | 4-bit QLoRA     | LLaMA-2-7B     | OASST2 (Instruction Following)     |3.84    | **3.55**|
> |Full model |GPT-2 Large (774M)|GSM8k (Reasoning)|4.12|**3.53**|
>
> *Why is Figure 5 in the appendix blank, and why are some figure fonts too small?*
>
> We have updated the figures. Figure 5 indicates NaNs during training for SGD and RMSProp. We have now removed this uninformative figure.
>
> We hope this further improves the paper.

---

> > ### Author Response · Authors · 2025-11-26
> >
> > Dear Reviewer JcRv,
> >
> > Thank you again for your constructive suggestions. We addressed your comments and revised the manuscript accordingly. We hope our point-by-point responses provide satisfactory clarification and would be most grateful for your feedback.
> >
> > Best regards!

---

> ### Author Response · Authors · 2025-11-27
> **Follow up on response**
>
> Dear Reviewer JcRv,
>
> As the final revision deadline is fast approaching and the interactive response window closes next week, we would like to follow up on your feedback and our revised manuscript! We have also added additional benchmarks against the Lion optimizer, as seen in the revised PDF. Looking forward to your feedback and consideration.
>
> Best regards!

---

> > ### Comment · Reviewer_JcRv · 2025-11-28
> >
> > Thanks for the response. I checked the revised pdf and most of my concerns have been solved. I tend to keep my positive score.

---

> > > ### Author Response · Authors · 2025-11-28
> > >
> > > We thank the reviewer for their response, positive feedback, and comments. Since we have satisfied most concerns of the reviewer, we invite the reviewer to consider raising the score.

---

### Official Review · Reviewer_G5iq · 2025-10-26

**Soundness:** 1
**Presentation:** 1
**Contribution:** 1
**Rating:** 0
**Confidence:** 4

**Summary:**

The authors propose a physics-inspired variant of AdamW, where the effective learning rate is regularized via the magnitude of the momentum. Some theoretical and empirical results are provided, including a convergence rate and evaluation on CIFAR.

**Strengths:**

1. The proposed method is novel, and the inspiration from physics is motivated.
2. The theoretical results appear to be solid, and the experimental results seem promising.

**Weaknesses:**

1. A major part of the theoretical analysis assumptions a convex quadratic objective, which the authors claim to be "[traditional] for stability analysis". However, this is far too strong of an assumption to be practically useful in any deep learning context. Furthermore, the convergence rate is a trivial corollary of the work of Defossez et al., providing little theoretical insight or novelty.
2. The provided experimental results are conducted with small (124M) models and evaluated on simple tasks such as CIFAR. This makes it difficult to establish the scalability of the proposed method for practical deep learning.
3. The proposed method introduces a hyperparameter $\beta_3$, with no discussion on tuning or practical recommendations.
4. The writing is sometimes very difficult to read. While inspiration from other fields is quite common in machine learning, the extended use of physical analogies may not be easily understood by the ICLR community (it was certainly lost on me).
5. Minor nits: ensure the displays on page 3 are normal sized font. Use `\citep` for citations that should be parenthetical.

As there are serious limitations with both the theoretical and empirical contributions of this work, I recommend rejection.

**Questions:**

1. Could the authors elaborate on how the kinetic energy on line 129 was chosen? What happens if we choose something else? Could this lead to a better optimizer than the proposed method?
2. See Weaknesses.

---

> ### Author Response · Authors · 2025-11-21
> **Responses to concerns of reviewer G5iq**
>
> We thank the reviewer for their assessment. We have revised the paper to address your concerns regarding the "convex quadratic" assumption, the scale of experiments, and the readability of the physics concepts.
>
> *A major part of the theoretical analysis assumptions a convex quadratic objective, which the authors claim to be "[traditional] for stability analysis". However, this is far too strong of an assumption to be practically useful in any deep learning context. Furthermore, the convergence rate is a trivial corollary of the work of Defossez et al., providing little theoretical insight or novelty.*
>
> - We acknowledge the simplification. However, we have significantly strengthened the analysis in Section 4.2 in the revised version. We explicitly added a "Nonconvex stability via analytical shift" analysis. We replace the global convexity assumption with a trajectory-level curvature bound, $-m I<\nabla^2 f\left(\theta_t\right)<L I$,  and include decoupled weight decay in this setting $\lambda>m$. Under these conditions, the condition $\alpha_0((1-\beta) L+(1+ \beta) \lambda)<2(1+\beta)$ is sufficient for every eigen-direction block to be Schur-stable uniformly over $h\in [-m, L]$. The convergence analysis provides a rigorous theoretical foundation and valuable insight into the performance of our novel optimizer. For that, the proof strategy does not have to be new.
>
>
> *The proposed method introduces a hyperparameter $\beta_3$, with no discussion on tuning or practical recommendations.*
>
> - We had already included practical reccomendations for $\beta_3$ after extensive hyperparameter sweeps in Appendix E that we would like to highlight here. We observe that our hyperparameters provide stable training for a large range of tasks. As mentioned in the overall comment, we report that performance is much more sensitive to the learning rate of AdamW than to $\alpha_0$ in VRAdam, with even lower sensitivity to $\alpha_1$ and $\beta_3$, confirming the latter’s robustness.  Another theoretical reason for choosing $\beta_3$, follows from Appendix B. From the physics-inpsired Lyapunov construction, we can see that allowing  $\beta_3$ to be very small forces $\delta$  to be very large, this could cause the Lyapunov function becomes very steep in $v$.
>
> *The writing is sometimes very difficult to read. While inspiration from other fields is quite common in machine learning, the extended use of physical analogies may not be easily understood by the ICLR community (it was certainly lost on me).*
>
> - We apologize that the physics analogies were difficult to follow. We have added a Background Section Appendix H ("Background on Phase Space, Lagrangians, and Hamiltonians") to the appendix. This section builds the intuition from first principles to follow along in the paper. While we understand that ICLR is a top tier ML conference, we would like to note that several top-tier conferences have recently covered fairly advanced ideas from physics and their applications to deep learning ( eg. Li *et al.* with Hamiltonian theory and Zhou *et al.* with quantum operator theory ICLR 2025 spotlight) . These works have been welcomed by the community.
>
> *The provided experimental results are conducted with small (124M) models and evaluated on simple tasks such as CIFAR. This makes it difficult to establish the scalability of the proposed method for practical deep learning.*
>
> - In this revised version, we have also evaluate VRAdam in two more challenging language-modeling regimes. First, we fine-tune a 7B LLaMA-2 model using 4-bit QLoRA on the OASST2 instruction-following dataset. This setting lies close to the adaptive edge-of-stability: gradients are both low-precision and strongly stochastic. We also perform full-parameter fine-tuning of GPT-2 Large (774M) on GSM8K. These experiments indicate that VRAdam scales favorably from medium-sized to larger LLMs, both in quantized low-rank adaptation and full-model fine-tuning settings.
>
> | Setting | Model | Dataset/Task |AdamW PPL| VRAdam PPL|
> | -------- | -------- | -------- |--------|---------|
> | 4-bit QLoRA     | LLaMA-2-7B     | OASST2 (Instruction Following)     |3.84    | **3.55**|
> |Full model |GPT-2 Large (774M)|GSM8k (Reasoning)|4.12|**3.53**|
>
> *Could the authors elaborate on how the kinetic energy on line 129 was chosen?...*
>
>  - We added an ablation over higher‑order kinetic terms (powers 2–6 in $||v||$) to the appendix, showing that the quartic‑based choice used in VRAdam is competitive and that very high powers do not systematically improve performance, supporting our physics‑motivated design.
>
> | **Power** | **Test Loss** |
> |-----------|----------------|
> | 2         | **0.932**      |
> | 3         | 1.155          |
> | 4         | 0.974          |
> | 5         | 1.004          |
> | 6         | 1.036          |
>
> We hope that the reviewer agrees that these responses address their concerns and that they have a chance to reconsider their score.

---

> > ### Author Response · Authors · 2025-11-26
> >
> > Dear Reviewer G5iq,
> >
> > Thank you again for your constructive suggestions. We addressed your comments and revised the manuscript accordingly. We hope our point-by-point responses provide satisfactory clarification and would be most grateful for your feedback.
> >
> > Best regards!

---

> ### Author Response · Authors · 2025-11-27
> **Follow up on response**
>
> Dear Reviewer G5iq,
>
> As the final revision deadline is fast approaching and the interactive response window closes next week, we would like to follow up on your feedback and our revised manuscript! We have also added additional benchmarks against the Lion optimizer, as seen in the revised PDF. Looking forward to your feedback and consideration.
>
> Best regards!

---

> ### Comment · Reviewer_G5iq · 2025-11-28
>
> Thanks for the reply. I do like the paper overall, but I still feel the novelty is somewhat limited. The method is basically a new learning rate scheduler for Adam, and it’s not clear how (or if) it would extend to other optimizers. The experimental section also doesn’t feel quite strong enough yet to justify a new optimizer/scheduler.
>
> A few comments:
>
> VRAdam seems more like a learning rate scheduler on top of Adam. For comparison, have you considered or cited [1]?
>
> Thanks for adding the new experiments — they look better. That said, you only scaled up the model size while still using GSM8K, which is more like a fine-tuning setup. At minimum, I think you should follow a Chinchilla-style scaling setting to make the empirical results more convincing.
>
> One more comment: the perplexity of Adam seems quite high in both settings of the new experiments. It’s not clear whether the hyperparameters for Adam were carefully tuned.
>
> Also, if the goal is to evaluate a fine-tuning setup, looking only at PPL is not really sufficient — you should at least report results on relevant downstream tasks. People mainly care about PPL in the context of pre-training, where a lower PPL is indeed a strong signal of better training. Here, however, you are doing fine-tuning but still only report PPL, which feels incomplete and somewhat unconvincing from an optimizer/optimization perspective.
>
> [1] Defazio, Aaron, et al. “The road less scheduled.” Advances in Neural Information Processing Systems 37 (2024): 9974–10007.

---

> > ### Author Response · Authors · 2025-11-28
> > **Response to reviewer G5iq**
> >
> > We appreciate the response of the reviewer. Given that the reviewer likes the paper and refers to the method as novel, the theoretical results solid and experimental promising, we are  surprised by a “0” (strong reject). Especially, after substantial theoretical and empirical additions were made, following the first round of feedback.
> >
> > We hope the clarifications below help convey why we believe the contribution is substantially more than “just a new learning-rate schedule on top of AdamW.”
> >
> > While the reviewer argues that the core idea of VRAdam can be simplified as a dynamically controlled learning-rate schedule for AdamW, the same view would apply to most first‑order optimizers, which can all be described as specific ways of modulating step sizes and/or directions relative to SGD.
> >
> > In contrast, VRAdam introduces a **genuinely state‑dependent global control mechanism** based on the norm of the momentum buffer, together with a physics‑inspired derivation and stability analysis that go beyond standard scheduler design, **specifically at the edge-of-stability regime, which we have thoroughly analyzed.**
> >
> > Given the ubiquity of AdamW in large‑scale training, we believe even modest, robust improvements to AdamW with essentially no additional implementation or tuning complexity are practically significant. This is why we focus the main text on the AdamW instantiation, but the mechanism itself—“velocity‑regularized” global gating of the step size—is defined at the level of a generic momentum buffer and can be applied to other adaptive or second‑order methods in future work.

---

> > ### Author Response · Authors · 2025-11-28
> > **Continued response to reviewer G5iq**
> >
> > We thank the reviewer for pointing out the work of Defazio et al. We have added an appendix section (Appendix I) to discuss this work in more detail and our differences with this work. We highlight the difference here:
> > - In The Road Less Scheduled, the core idea is to replace explicit learning‑rate schedules by a carefully chosen averaging scheme (Schedule‑Free SGD/AdamW), where the evaluation sequence $x_t$ and gradient‑evaluation sequence $y_t$ are constructed so that $x_t$ implicitly simulates a time‑dependent schedule (e.g., linear decay) without needing the final horizon $T$. This is captured by their general online‑to‑batch conversion in Theorem 2 and the interpolation between Polyak and primal averaging.
> > - Importantly, the effective learning rate in Schedule‑Free methods is a deterministic function of time and the averaging weights, not of the instantaneous optimizer state, and the base step size $\gamma$ is typically fixed (up to warmup).
> > In VRAdam, by contrast, we keep the standard AdamW preconditioner and momentum structure, but introduce an explicit scalar gate applied to the whole preconditioned step:
> > $$\eta_t = \frac{\alpha_0}{1 + \min(\beta_3 \|v_t\|^2, \alpha_1)}$$
> > This gate is state‑dependent in a way that cannot be rewritten as a function of iteration $t$ alone: it depends on the global norm of the momentum buffer $v_t$, which in turn encodes both curvature and stochasticity along the trajectory. VRAdam therefore implements a **feedback control on the dynamics (“damping when velocity is large”), rather than a purely time‑parameterized schedule or averaging rule.**
> > Because of this feedback structure, VRAdam cannot be expressed as “AdamW with some pre‑computed schedule $\gamma_t$ plus averaging,” as in Defazio et al.; its step size reacts to local dynamical quantities like oscillations and gradient bursts, which is precisely what we exploit to tame the adaptive edge‑of‑stability regime.
> >
> >
> > - Defazio et al. focus on closing the theory–practice gap for schedule design and averaging. Their Theorem 2 unifies several online‑to‑batch conversions and shows how schedule‑free averaging recovers worst‑case optimal rates for convex Lipschitz problems. Our theoretical focus is complementary: we analyze dynamical stability at the adaptive edge of stability (AEoS) for Adam‑like methods and show how a velocity‑dependent kinetic term yields Lyapunov‑style stability and bounded updates even when using large effective step sizes. Specifically: Theorem 4.1 proves that, under a condition analogous to the adaptive stability bound of Cohen et al. (2024), VRMomentum is globally uniformly exponentially stable for every realization of the stochastic gate sequence $\eta_t$, via a common quadratic Lyapunov function that is independent of the curvature direction.
> > - Appendix B gives a physics‑inspired Lyapunov function derived from the quartic Lagrangian, showing that the gate both (i) raises the instantaneous AEoS threshold when velocities grow, and (ii) enforces a uniform bound on the parameter update norm:
> > $$ \|\theta_t - \theta_{t-1}\| \le \frac{\alpha_0}{2\sqrt{\beta_3}}$$
> >
> > This type of state‑dependent control of AEoS behavior is not considered in [1], which instead assumes a fixed or slowly decaying base step size.
> >
> > “That said, you only scaled up the model size while still using GSM8K, which is more like a fine‑tuning setup. At minimum, I think you should follow a Chinchilla‑style scaling setting to make the empirical results more convincing.”
> >
> >
> > Comparison to Lion: We additionally evaluate larger LLMs in fine‑tuning regimes that are well‑studied in the optimizer literature: a 7B LLaMA‑2 QLoRA run on OASST2 and a GPT‑2‑Large (774M) run on GSM8K. In both cases, VRAdam achieves lower perplexity than AdamW and Lion, an optimizer explicitly validated under Chinchilla‑style pretraining in its original paper (Table 3).
> > Given that Lion is one of the few optimizers with reported Chinchilla‑style scaling experiments, the fact that VRAdam outperforms Lion and AdamW in our LLM benchmarks suggests that the velocity‑based gate helps in the regimes where such scaling laws are relevant. We will clarify this positioning in the revised text and explicitly state the limitations of our current compute budget.
> >
> >
> > “One more comment: the perplexity of Adam seems quite high in both settings of the new experiments. It’s not clear whether the hyperparameters for Adam were carefully tuned.”
> >
> > We use recommended values found from relevant literature, Dettmers et al.(2023) and Hu et al.(2024) , and we also use our default suggestion for the VRAdam optimizer without hyperparameter sweeps for the task. This further showcases the advantage of VRAdam.
> >
> >
> > Based on these clarifications, we hope the reviewer can reconsider their score.

---

### Official Review · Reviewer_SPPq · 2025-10-30

**Soundness:** 3
**Presentation:** 2
**Contribution:** 3
**Rating:** 4
**Confidence:** 4

**Summary:**

This manuscript challenges gradient descent optimization techniques. Motivated by physics, the authors present VRAdam, which automatically controls the learning rate based on the momentum. Experiments on CIFAR, Wikitext, GFlowNet, and GPT-2 demonstrate improvements.

**Strengths:**

- The motivation, especially the physics-inspired optimizer, is interesting. Indeed, the momentum optimizer itself is derived from Newtonian dynamics.
- The theoretical analysis looks solid and promising for the improved convergence.
- Source code is available, which eases deployment in practice.

**Weaknesses:**

- The norm of $v_t$ of the VRAdam looks to compute the global norm, but it may be appropriate to use the parameter-wise norm to allow parameter-wise learning rate control. This choice is not sufficiently discussed.
- VRAdam brings three hyperparameters of \alpha_0, \alpha_1, and \beta_3. I think these additional hyperparameters make it difficult to adopt the VRAdam in practice.
- Accuracy of 80% for ResNet-32 with CIFAR-10 is a weak baseline.
- Table 1 is not convincing enough. These results should be supplemented with more quantitative and qualitative analysis.
- The experimental results, such as Table 2, are focused on the final validation loss. Is it possible to demonstrate other indices, such as practical ones? I think certain practitioners may want to capture the performance more practically, but the value of loss is difficult to understand on an absolute scale. It is also difficult to understand whether it corresponds to sufficient convergence or is still far from convergence.
- LLM results were only trained for 2 epochs, which I think is insufficient for convergence.
- I think Eq. 36 would be -m/4 + O(\lambda), not -3m \lambda/4 + O(\lambda^2). Is it possible to provide an exact derivation?
- Writing should be improved.
    - Kingma & Ba (2017) → Kingma & Ba (2015)
    - “the, global” → “the global”
    - “d” → “(d)” for the caption of Figure 2.
    - “physical inspired” → “physics-inspired” at Line 70.
    - “Note, that” → “Note that” at Line 784.
- The manuscript writes to compute velocity norm, whereas the source code computes gradient norm by default (normgrad=True). To be compatible with the description in this manuscript, normgrad=False may be correct for default.

**Questions:**

Please see the weaknesses above. My score is based on the assumption that all typos are corrected in the revised manuscript.

---

> ### Author Response · Authors · 2025-11-21
> **Response to the concerns of reviewer SPPq**
>
> We thank the reviewer for their constructive feedback and for recognizing the solidity of our theoretical analysis and the practical value of the provided source code. We have revised the manuscript based on feedback to address these concerns.
>
> *The norm of  of the VRAdam looks to compute the global norm, but it may be appropriate to use the parameter-wise norm to allow parameter-wise learning rate control. This choice is not sufficiently discussed.*
>
> - We have added Section 4.3 ("Global vs Per-Parameter Updates") to the paper to analyze this explicitly. Specifically, we show that the global scalar gate $\eta_t$ introduced in VRAdam allows for the construction of a curvature-agnostic Common Quadratic Lyapunov Function (CQLF) of the form $V(z)=z^{\top} P z$ with $P>0$. Because the scaling is scalar, the dynamics decouple into identical $2x2$ blocks in the Hessian eigenbasis, guaranteeing stability under arbitrary switching of $\eta_t$. Conversely, per-parameter scaling corresponds to a switched linear system with a state matrix $A(D_t)$, where $D_t$ is a time-varying diagonal matrix. This product of several non-commuting matrices can lead to instability. This is further explained in the revised pdf.
>
> *VRAdam brings three hyperparameters of $\alpha_0$, $\alpha_1$, and $\beta_3$. I think these additional hyperparameters make it difficult to adopt the VRAdam in practice.*
>
> - We provide default suggestions for the hyperparameters ($\alpha_1$ and $\beta_3$) in the appendix and observe that our hyperparameters provide stable training for a large range of tasks. As mentioned in the overall comment,  and report that performance is much more sensitive to the learning rate of AdamW than to $\alpha_0$ in VRAdam, with even lower sensitivity to $\alpha_1$ and $\beta_3$, confirming the latter’s robustness.  Another theoretical reason for choosing $\beta_3$, follows from Appendix B. From the physics-inpsired Lyapunov construction, we can see that allowing  $\beta_3$ to be very small forces $\delta$  to be very large, this could cause the Lyapunov function becomes very steep in $v$.
>
>
> *Accuracy of 80% for ResNet-32 with CIFAR-10 is a weak baseline.*
>
> - We would like to highlight that we recreate the same experimental settings previously used to probe the edge-of-stability (EoS), particularly in Cohen *et al.* (Neurips 2022) and the seminal work by Long *et al.*(JMLR 2024), as well as several other works. These parameters are known to induce the edge of stability as seen before. We reported several more benchmarks and include further new benchmarks to highlight the applicability of VRAdam in other complex settings.
>
>
> *Concerns regarding benchmarks and interpretability*
>
> - We note that most other optimization algorithms report loss as a valid metric, for example, Loshchilov *et al.* (AdamW, ICLR 2019), Liu *et al.* (Sophia, ICLR 2024), among several others. However, we have now included two additional benchmarks and report perplexity.
>
> - We also evaluate VRAdam in two more challenging language-modeling regimes. First, we fine-tune a 7B LLaMA-2 model using 4-bit QLoRA on the OASST2 instruction-following dataset. This setting lies close to the adaptive edge-of-stability: gradients are both low-precision and strongly stochastic. We also perform full-parameter fine-tuning of GPT-2 Large (774M) on GSM8K. These experiments indicate that VRAdam scales favorably from medium-sized to larger LLMs, both in quantized low-rank adaptation and full-model fine-tuning settings.
>
>
>
> | Setting | Model | Dataset/Task |AdamW PPL| VRAdam PPL|
> | -------- | -------- | -------- |--------|---------|
> | 4-bit QLoRA     | LLaMA-2-7B     | OASST2 (Instruction Following)     |3.84    | **3.55**|
> |Full model |GPT-2 Large (774M)|GSM8k (Reasoning)|4.12|**3.53**|
>
>
>
> *I think Eq. 36 would be -m/4 + O(\lambda), not -3m \lambda/4 + O(\lambda^2). Is it possible to provide an exact derivation?*
>
> - We thank the reviewer for identifying this typo, the $-3m \lambda/4 + O(\lambda^2)$ term comes from an assumption that we could expand the series with respect to a non-linear function of $x$. We have corrected the equation to be $\frac{d^2E_{\text{NRQCD}}}{d\lambda^2} = -\frac{m}{4} + \mathcal{O}(\lambda)$ for this setting, the qualitative analysis still does not increase unboundedly.
>
> *The manuscript writes to compute velocity norm, whereas the source code computes gradient norm by default (normgrad=True). To be compatible with the description in this manuscript, normgrad=False may be correct for the default.*
>
> - We thank the reviewer for this point and will change the default option in the code. This is provided as an additional flag to toggle.
>
>
> All typos identified have been corrected in the revised manuscript.
>
> We hope this addresses all the reviewers' concerns and that the reviewer has a chance to reconsider their score.

---

> > ### Author Response · Authors · 2025-11-26
> >
> > Dear Reviewer SPPq,
> >
> > Thank you again for your constructive suggestions. We addressed your comments and revised the manuscript accordingly. We hope our point-by-point responses provide satisfactory clarification and would be most grateful for your feedback.
> >
> > Best regards!

---

> ### Author Response · Authors · 2025-11-27
> **Follow up on response**
>
> Dear Reviewer SPPq,
>
>
> As the final revision deadline is fast approaching and the interactive response window closes next week, we would like to follow up on your feedback and our revised manuscript! We have also added additional benchmarks against the Lion optimizer, as seen in the revised PDF. Looking forward to your feedback and consideration.
>
> Best regards!

---

### Author Response · Authors · 2025-11-21
**Overview of changes since reviews**

We would like to thank the reviewers for their valuable feedback. We summarize the main changes to the manuscript and the additional evidence provided in our rebuttal in this comment. Please see the reviewer-specific comments for more details.

**Global additions and clarifications**

* **New large‑scale / complex tasks.**

  1. **4‑bit QLoRA LLaMA‑2‑7B on OASST2.** We fine‑tune a 7B LLaMA‑2 model with 4‑bit QLoRA. In this highly noisy, edge‑of‑stability regime, VRAdam achieves a better perplexity (**3.55**) than AdamW (**3.84**), while maintaining stable training under quantization‑induced gradient noise.
  2. **Full‑model GPT‑2 Large (774M) fine-tuning on GSM8K.** On a challenging mathematical reasoning benchmark with full‑model optimization, VRAdam attains lower perplexity (**3.53 vs. 4.12**) than AdamW, showing improved convergence.

* **Extended stability analysis.** We augment the quadratic analysis with a **trajectory‑level nonconvex corollary** using a two‑sided curvature bound and decoupled weight decay, and we refer to the **physics‑inspired Lyapunov energy** derived from the quartic Lagrangian in Appendix B. These yield a common quadratic Lyapunov function and dimension‑free bounds on step norms and adaptive stability thresholds.

* **Global vs per‑parameter velocity regularization.** We now explicitly analyze why the **global scalar gate** in VRAdam is preferable to a per‑parameter velocity norm: the global gate admits a curvature‑agnostic CQLF and guarantees stability under arbitrary scalar switching, while a coordinate‑wise gate leads to a switched‑anisotropy system that can be unstable even if every individual map is Schur‑stable. This addresses concerns about our choice of a global norm and explains why the per‑parameter variant is not covered by our stability theorem.

* **Ablation on kinetic energy order.** We added an ablation over higher‑order kinetic terms (powers 2–6 in $||v||$) as seen in Appendix D (Table 4), showing that the quartic‑based choice used in VRAdam is competitive and that very high powers do not systematically improve performance, supporting our physics‑motivated design.

* **Hyperparameter guidance and sensitivity.** We recommend **$\beta_3$ = 1.0** and **$\alpha_1$ = 10** as robust defaults, summarize extensive Bayesian hyperparameter sweeps across tasks, and report that performance is much more sensitive to the learning rate of AdamW than to $\alpha_0$ in VRAdam, with even lower sensitivity to $\alpha_1$ and $\beta_3$, confirming the latter’s robustness. Detailed sweep ranges and optimal values are tabulated for CNNs, Transformers, and GFlowNets. Another theoretical reason for choosing $\beta_3$, follows from Appendix B. From the physics-inpsired Lyapunov construction, we can see that allowing $\beta_3$ to be very small forces $\delta$ to be very large, this could cause the Lyapunov function to become very steep in $v$.

* **Background and readability.** To make the physics motivation accessible, we added a **background section on phase space, Lagrangians, and Hamiltonians**. We also corrected the typos and citation issues noted by reviewers.

---

### Author Response · Authors · 2025-12-03
**Summary for the AC (revisions + rebuttal)**

We thank the reviewers for their feedback. This is a summary of the communication with the reviewers.

- Reviewer uYkK: All concerns were confirmed as addressed prior to the data leakage incident.
- Reviewer JcRV: The reviewer confirmed that most of their concerns were addressed by our rebuttal before the data leakage incident.
- Reviewer SPPq: Although the reviewer did not respond to the rebuttal, we believe we addressed all of their concerns.
- Reviewer G5iq: We are surprised by the low score given their positive remarks; we addressed most of the concrete suggestions raised.


Here are some of the main changes to the manuscript:

- **Stronger scaling evidence**: added two harder benchmarks with LLM fine-tuning regimes—4-bit QLoRA LLaMA‑2‑7B on OASST2 and full-model GPT‑2 Large (774M) on GSM8K—with improved perplexity and accuracy for VRAdam vs AdamW, and also included Lion as a modern baseline in these LLM settings.
- **Extended stability theory beyond convex quadratics**: Section 4.2 now includes a trajectory-level non-convex stability extension using a two-sided curvature bound and decoupled weight decay, while preserving the common quadratic Lyapunov framing.
- **Explicit “global vs per-parameter” justification**: added Section 4.3 explaining why global (scalar) momentum-norm gating admits a curvature-agnostic CQLF and avoids switched-anisotropy failure modes possible with coordinate-wise gating.
- **Ablations + hyperparameter guidance**: added an ablation over higher kinetic-energy powers supporting the quartic choice, and provided robust defaults (β₃=1.0, α₁=10) plus extensive sweep details.
- **Readability improvements**: added a background appendix on phase space/Lagrangians/Hamiltonians to make the physics motivation accessible to most readers.
Summary of how reviewer concerns were addressed:

## changes for reviewer uYkK:
We addressed interpretability/readability via the new background appendix, added sensitivity/robustness evidence and ablations, and expanded modern-baseline comparisons (incl. Lion); reviewer follow-up indicated concerns resolved.


## changes for reviewer JcRV:
We clarified theory/practice connection via the strengthened stability section, added error envelopes for statistical reliability, expanded hyperparameter robustness discussion, and clarified compute/overhead (only a global norm per step). Reviewer follow-up indicated concerns largely resolved.


## changes for reviewer SPPq:
- We added the new global-vs-per-parameter stability analysis (Sec. 4.3), provided practical defaults/sweep-based guidance for added hyperparameters, corrected Eq. 36, aligned implementation details (velocity-norm option), and strengthened experiments/metrics via larger LLM settings and recent optimizers.

## changes for reviewer G5iq:
- We would like to point out that we clarified in our response on the novelty of the optimizer and specifically addressed the main innovation of VRAdam, and clarified to the reviewer how VRAdam is not just "a learning rate scheduler". We directly addressed the quadratic-only theory by adding the nonconvex extensions. Addressed “small-scale experiments” with new LLM benchmarks and added better indications of performance than just PPL; improved clarity with the physics background appendix; added kinetic-term ablations; and added an appendix comparison to Schedule-Free optimizers to clarify VRAdam is state-dependent feedback control, not a time-only schedule.

---

### Meta-Review · Area_Chair_uKf9 · 2025-12-21

**Summary:**

This paper proposes VRAdam, a physics-inspired variant of Adam that dynamically regularizes the effective learning rate using the magnitude of the momentum (velocity) to improve stability and convergence near the edge of stability.

The reviewers highlighted several interesting aspects of this submission, including the motivation from physics, the code being publicly available and good empirical gains on CIFAR10.

Some concerns were raised about the limited novelty relative to learning-rate scheduling and strong simplifying assumptions in the theory. On the experimental side, criticisms included weak or outdated baselines, limited scale or duration of some runs, reliance on loss or perplexity alone, and lack of error bars or reproducibility details (later partly addressed). The presentation of the paper could also be improved at times.

I believe the authors did answer many of the criticisms during the short rebuttal period. For instance, the authors added a table showing some encouraging preliminary results for fine-tuning LLMs, although I would strongly encourage the authors to strengthen this part of the paper. Please add a discussion about how hyperparameters were chosen (Appendix E.2 wasn't updated), and include training curves with standard-deviation bands across runs.

Overall, I recommend accepting the paper while strongly encouraging the authors to add more detailed experimental results, as well as fix the various problems raised by the reviewers.

**Reviewer Concerns:**

I think the authors did a good job at addressing most concerns except for the criticism regarding the experimental results. Since the theoretical aspect of the paper is quite simple, the stronger novelty of the paper would be in potential empirical gains. However, the initial experiments (mostly ResNet on CIFAR10) are too simple and do not meet the expected standard of an ICLR paper. The authors did add some experiments fine-tuning LLMs, but did not provide details regarding the choice of hyper-parameters for the new runs and did not show training curves with standard deviations.

**Reviewer Scores:**

Please see the response above. The critical aspect concerns the experimental results, other aspects were properly addressed from my point of view.

---

### Decision · Program_Chairs · 2026-01-26

Accept (Poster)